# Leave No Stone Unturned: Mine Extra Knowledge for Imbalanced Facial Expression Recognition

**Yuhang Zhang, Yaqi Li, Lixiong Qin, Xuannan Liu, Weihong Deng**
Beijing University of Posts and Telecommunications
{zyhzyh, yaqili, lixiongqin, xuannanliu, whdeng}@bupt.edu.cn

## Abstract

Facial expression data is characterized by a significant imbalance, with most collected data showing happy or neutral expressions and fewer instances of fear or disgust. This imbalance poses challenges to facial expression recognition (FER) models, hindering their ability to fully understand various human emotional states. Existing FER methods typically report overall accuracy on highly imbalanced test sets but exhibit low performance in terms of the mean accuracy across all expression classes. In this paper, our aim is to address the imbalanced FER problem. Existing methods primarily focus on learning knowledge of minor classes solely from minor-class samples. However, we propose a novel approach to extract extra knowledge related to the minor classes from both major and minor class samples. *Our motivation stems from the belief that FER resembles a distribution learning task, wherein a sample may contain information about multiple classes. For instance, a sample from the major class surprise might also contain useful features of the minor class fear.* Inspired by that, we propose a novel method that leverages re-balanced attention maps to regularize the model, enabling it to extract transformation invariant information about the minor classes from all training samples. Additionally, we introduce re-balanced smooth labels to regulate the cross-entropy loss, guiding the model to pay more attention to the minor classes by utilizing the extra information regarding the label distribution of the imbalanced training data. Extensive experiments on different datasets and backbones show that the two proposed modules work together to regularize the model and achieve state-of-the-art performance under the imbalanced FER task. Code is available at https://github.com/zyh-uaiaaaa.

## 1 Introduction

Facial expression recognition (FER) plays a crucial role in enabling machines to understand human emotional states, thus facilitating the realization of machine intelligence [18, 11, 25]. Recent research efforts [42, 35, 49] have employed in-the-wild datasets to train FER models, resulting in impressive performance in terms of overall classification accuracy. However, these FER datasets suffer from a significant imbalance, with a higher abundance of happy and neutral expression faces, which are easily obtainable from the internet and daily life, compared to faces displaying negative expressions such as fear or disgust [6, 32]. This imbalance can have adverse effects on human-computer interaction when FER models misinterpret infrequently occurring negative emotions as frequently occurring positive emotions.

In this paper, our objective is to investigate imbalanced learning in the context of FER. We observe that existing FER methods [34, 43, 42, 29] yield relatively low performance in terms of the mean accuracy across all classes, mainly due to the highly imbalanced nature of the training data. While imbalanced (long-tailed) learning methods in the image classification domain have shown some

37th Conference on Neural Information Processing Systems (NeurIPS 2023).

improvements [52, 5], they provide limited enhancements in the mean accuracy within the FER field. This can be attributed to several factors. Firstly, FER datasets typically contain a small number of classes, resulting in marginal improvements in the mean accuracy of all classes. Secondly, the increase in accuracy for minor classes often comes at the expense of decreased accuracy for major classes. This imbalance disproportionately affects the overall accuracy, particularly when evaluating on imbalanced test sets.

To address the aforementioned problem, our objective is to design a method that can maintain high performance on major classes while improving the performance on minor classes to the greatest extent possible. We refrain from modifying the dataset structure through under-sampling or over-sampling, as these approaches may enhance the performance of minor classes at the expense of degrading the performance on major classes. Instead, we aim to leverage each sample fully to extract additional knowledge relevant to minor classes. *Our motivation is that FER resembles a label distribution learning task, implying that samples from major classes may contain information pertaining to minor classes as well.* For instance, the major class "surprise" shares certain similarities with the minor class "fear".

Inspired by that, we propose a novel method that mines extra knowledge regarding minor classes from samples belonging to both major and minor classes. This extra information enhances the recognition performance on minor classes without significantly compromising the performance on major classes. Specifically, we employ the concept of attention map consistency [7], which is utilized in the EAC method [50] to prevent FER models from memorizing noisy labels. We observe that EAC employs attention maps from all expression classes, rather than just the labeled class, to regularize a specific sample, resembling the idea of label distribution learning. Building upon this insight, we introduce a re-balanced attention consistency module to address the imbalanced FER task for the first time. In our approach, attention maps for all expression classes can be extracted from a given FER sample. We introduce re-balanced weights, following the design of [5], which are set inversely proportional to the effective number of samples belonging to each class. These weights facilitate the model in extracting additional knowledge related to minor classes from all training samples. Furthermore, as the classification loss may still be affected by the imbalanced data, we propose a re-balanced smooth labels module that utilizes the acquired re-balanced weights and the extra knowledge of the imbalanced label distribution. This module guides the model to focus more on minor classes during the decision-making process.

To demonstrate the effectiveness of our proposed method, we conducted extensive experiments on different FER datasets, including datasets with varying imbalance factors [5]. Ablation studies confirmed that each proposed module contributes to the enhancement of mean accuracy, and the two modules synergistically achieve state-of-the-art performance. Moreover, we evaluated the generalization ability of our method by combining it with different backbones including transformers. The experimental results showcased that our method is lightweight, easy to implement, and seamlessly compatible with various backbones, thus significantly improving their performance.

The main contributions of our work are listed as follows:

- We highlight the imbalanced learning problem in FER and find existing methods in the large-scale image classification field might fall short in the FER field, we benchmark existing FER methods utilizing both overall accuracy and mean accuracy on all classes.

- We propose a novel method consisting of two modules: re-balanced attention consistency (RAC) and re-balanced smooth labels (RSL). RAC mines extra knowledge pertaining to minor classes from both major and minor samples, thereby enhancing performance on minor classes without compromising performance on major classes. RSL leverages the imbalanced label distribution to further regularize the classification loss and prioritize the decision-making process for minor classes.

- Our proposed method is easy to implement, lightweight, and seamlessly compatible with different backbone architectures including transformers. Through extensive experiments, we validate that our proposed method achieves state-of-the-art performance in terms of both overall accuracy and mean accuracy of all classes across different FER datasets and backbone architectures.

## 2   Related work

**Facial expression recognition** methods in recent years primarily focus on extracting effective features from in-the-wild datasets using deep learning models [22, 46, 13, 16, 20, 24, 21, 23]. In-the-wild datasets [26, 6] pose greater challenges and are more prone to noise due to label inaccuracies, difficult poses, occlusions, and low-quality images compared to laboratory collected datasets [30, 14, 40]. These factors adversely affect the recognition performance of facial expression recognition (FER) models. Several works have addressed the real-world FER task. SCN [42] introduces a learnable temperature and a relabel module to weight FER samples and suppress noise. DMUE [37] learns sub-modules for each class to handle noisy labels. RUL [49] treats expression uncertainty as a relative concept and learns uncertainty values for images through comparisons, aiding overall feature learning. TransFER [45] pioneers the use of vision transformers for in-the-wild FER. EAC [50] employs erasing attention map consistency to prevent FER models from memorizing noisy labels in real-world FER datasets.

While these approaches achieve impressive overall accuracy on test sets, they tend to exhibit lower performance in terms of mean accuracy of all classes, particularly for minor classes. In fact, previous works primarily focus on mitigating noise in in-the-wild datasets, often overlooking the imbalanced nature of FER datasets. In this work, our objective is to address imbalanced learning in in-the-wild FER datasets and provide an orthogonal supplement for previous works.

**Imbalanced learning** in real-life datasets, where certain classes have a majority of samples while others have very few, has been addressed through three perspectives: data pre-processing, loss re-weighting, and model ensemble. This paper primarily focuses on the first two categories, as model ensemble methods [44, 48] are computationally intensive. Data pre-processing methods commonly involve data re-sampling [31, 38]. However, studies [15, 52] have shown that data re-sampling can negatively impact representation learning. Over-sampling may introduce duplicated samples, leading to the increased risk of overfitting, while under-sampling may discard valuable examples. Another technique is data augmentation [4, 53], which applies predefined transformations to augment the dataset, particularly for minority classes. However, finding effective augmentation methods for facial expression recognition (FER) data, which contain specific local features related to expressions, can be challenging. Loss re-weighting methods assign different weights to classes or instances during training, known as loss re-weighting [27, 39, 41, 5]. The aim is to propagate appropriate gradient values for all classes during training.

## 3   Method

In this section, we present a novel method for imbalanced facial expression recognition. Our approach focuses on extracting extra information of minor classes from both minor and major class samples, rather than solely relying on minor class samples. We introduce two modules, namely re-balanced attention consistency (RAC) and re-balanced smooth labels (RSL), to guide the model to extract balanced information from the entire training set.

### 3.1   Re-balanced attention consistency

We first introduce attention map consistency [7], which regularizes the attention maps to follow the same spatial transform if the input images are spatially transformed, which can help the model to learn transformation invariant features. EAC designs an imbalanced framework to utilize attention map consistency to prevent the models from memorizing the noisy labels. We find that instead of only extracting attention maps of the latent truth like utilizing GradCAM [36], EAC uses attention maps of all classes to regularize the FER model. We speculate that the reason lies in that utilizing attention maps of all classes is similar to label distribution learning [3, 51, 19], which mines information of several classes from one training sample. Inspired by that, we adapt attention map consistency to solve the imbalanced learning problem of FER for the first time. To be more specific, given a sample, the attention maps of all expression classes should follow the same spatial transform if the given sample is spatially transformed. Thus, the model could mine useful minor-class information from all samples instead of only the samples from the minor classes. Furthermore, we introduce the re-balanced attention consistency to regularize the model to focus more on the minor classes to achieve balanced learning. We set different weights to the attention maps of different classes

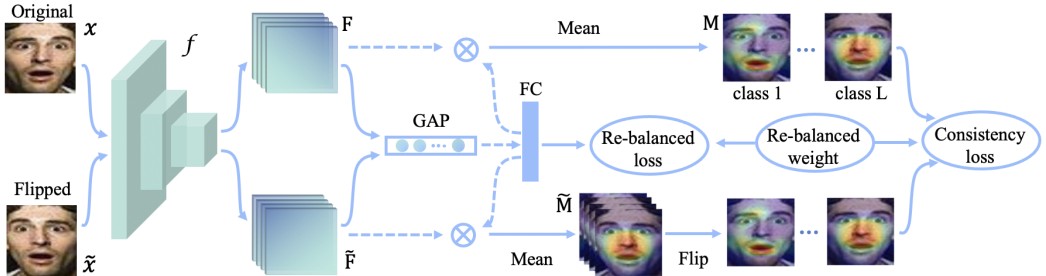

Figure 1: The illustration of re-balanced attention consistency. We propose re-balanced attention consistency to facilitate the model to mine extra transformation invariant knowledge of minor classes from both major and minor-class samples, which boosts the classification accuracy on minor classes while do not degrade the high accuracy on major classes.

and guide the model to learn more invariant information related to minor classes. Inspired by [5], which designs a re-weighting scheme that uses the effective number of samples of each class to re-balance the classification loss, we propose to use the effective number of samples to inversely the weigh attention maps of different classes. In the following, we formulate the re-balanced attention consistency module in detail.

Given images $\mathbf{x}$, we first flip them to get their flipped counterparts $\widetilde{\mathbf{x}}$. The features of $\mathbf{x}$ and $\widetilde{\mathbf{x}}$ are extracted from the last convolutional layer, denoted as $\mathbf{F} \in \mathbb{R}^{N \times C \times H \times W}$ and $\widetilde{\mathbf{F}} \in \mathbb{R}^{N \times C \times H \times W}$, where $N$, $C$, $H$ and $W$ respectively represent the number of images, channels, height and width. We input $\mathbf{F}$ and $\widetilde{\mathbf{F}}$ to the global average pooling (GAP) layer to get features $\mathbf{f} \in \mathbb{R}^{N \times C \times 1 \times 1}$ and $\widetilde{\mathbf{f}} \in \mathbb{R}^{N \times C \times 1 \times 1}$, and then resize them to $N \times C$. The classification loss is computed according to

$$l_{cls} = -\frac{1}{N} \sum_{i=1}^{N} (\log(\frac{e^{\mathbf{W}_{y_i}\mathbf{f}_i}}{\sum_{l=1}^{L} e^{\mathbf{W}_l\mathbf{f}_i}}) + \log(\frac{e^{\mathbf{W}_{y_i}\widetilde{\mathbf{f}}_i}}{\sum_{l=1}^{L} e^{\mathbf{W}_l\widetilde{\mathbf{f}}_i}})), \tag{1}$$

where $\mathbf{W}_{y_i}$ is the $y_i$-th weight of the fully connected (FC) layer and $y_i$ is the label of $\mathbf{x}_i$, $L$ is the total number of expression classes. CAM [53] is utilized to compute attention maps $\mathbf{A} \in \mathbb{R}^{N \times L \times H \times W}$ and $\widetilde{\mathbf{A}} \in \mathbb{R}^{N \times L \times H \times W}$ for $\mathbf{x}$ and $\widetilde{\mathbf{x}}$ following

$$A(i, l, h, w) = \sum_{c=1}^{C} W(l, c)F(i, c, h, w), \tag{2}$$

where $i, l, c, h, w$ represent the sample, expression class, channel, height and width number. We then re-balance the attention maps $\mathbf{A}$ and $\widetilde{\mathbf{A}}$ through weight $\mathbf{B} \in \mathbb{R}^{L}$ to get the re-balanced attention maps $\mathbf{M}$ and $\widetilde{\mathbf{M}}$ following

$$M(i, l, h, w) = B_l \cdot A(i, l, h, w). \tag{3}$$

The balance weight $\mathbf{B} \in \mathbb{R}^{L}$ is enlightened by the re-balanced weight based on the effective number in [5], which is computed following

$$B_l = \frac{1 - \beta}{1 - \beta^{n_l}}, \tag{4}$$

where $n_l$ is the number of training samples in class $l$, $\beta \in [0, 1)$ is the hyperparameter controlling the re-balanced weight, a larger value of $\beta$ emphasizes more of the minor samples, following [5], we set the $\beta$ as 0.9999 across all our experiments. As the attention map before and after the flip transformation should be consistent with each other, we compute the consistency loss between $\mathbf{M}$ and $Flip(\widetilde{\mathbf{M}})$ following

$$l_{cons} = \frac{1}{NLHW} \sum_{i=1}^{N} \sum_{l=1}^{L} \sum_{h=1}^{H} \sum_{w=1}^{W} ||M(i, l, h, w) - Flip(\widetilde{M})(i, l, h, w)||_2. \tag{5}$$

The training loss is computed as

$$l_{train} = l_{cls} + \lambda l_{cons}, \tag{6}$$

where $\lambda$ is the weight of the attention consistency loss, which determines the relative importance of the consistency loss compared to the classification loss in the overall training objective.

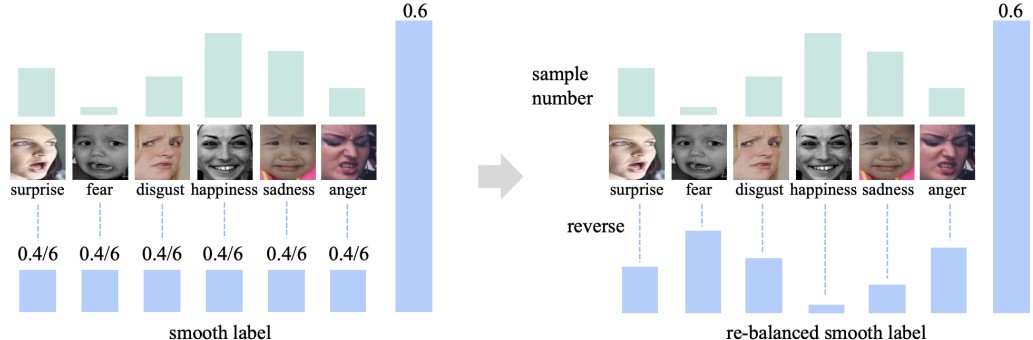

Figure 2: The illustration of re-balanced smooth labels. We propose re-balanced smooth labels to utilize the existing prior knowledge about the label distribution of the training set to guide the model towards placing more emphasis on minor classes during decision-making, while maintaining high performance on major classes.

## 3.2 Re-balanced smooth labels

In Section 3.1, we introduced the re-balanced attention consistency module, which enables the model to extract balanced and transformation invariant knowledge of minor classes from all training samples, resulting in improved classification accuracy for those classes. However, the imbalanced training set can still negatively impact the classification loss, denoted as $l_{cls}$. To address this, we present the re-balanced smooth labels module in this section. This module aims to regulate the classification loss and promote balanced learning.

We denote the prediction of the FER model towards $\mathbf{x}_i$ as $\mathbf{p}_i$, where $\mathbf{p}_i$ is the likelihood the model assigns to the $i$-th given sample $\mathbf{x}_i$, the one-hot label of $\mathbf{x}_i$ is denoted as $\mathbf{y}_i$. For simplicity, we re-write the classification loss towards $\mathbf{x}_i$ as Eq. 7 and things are the same with $\widetilde{\mathbf{x}}_i$.

$$l_{cls} = -\frac{1}{N}\sum_{i=1}^{N}H(\mathbf{y}_i, \mathbf{p}_i) = -\frac{1}{N}\sum_{i=1}^{N}\sum_{l=1}^{L}y(i,l)log(p(i,l)), \tag{7}$$

where $y(i,l)$ is "1" for the correct class of $\mathbf{x}_i$ and "0" for the rest. For a FER model trained with a re-balanced smooth label of parameter $\alpha$, we minimize instead the cross-entropy between the re-balanced targets $\widetilde{\mathbf{y}}_i$ and the model's outputs $\mathbf{p}_i$, where

$$\widetilde{y}(i,l) = (1-\alpha)y(i,l) + \alpha B_l/L, \tag{8}$$

$B_l$ is the re-balanced weight of the class $l$ given in Section 3.1.

## 3.3 Discussion with existing works

Our most related work is EAC. Though both methods utilize attention map consistency, EAC aims to address the noisy label learning task, while our method focuses on imbalanced learning. Furthermore, the motivations of the two works are different. EAC utilizes the difference of attention maps before and after the transformation to prevent the model to memorize noisy labels, while we propose the re-balanced attention consistency to guide the model to extract useful features related to minor classes from both major and minor samples. We novelly solve the imbalanced FER task through a label distribution learning perspective. We also propose a re-balanced smooth labels module to further regularize the classification loss from the negative effect of the imbalanced training set. We use the extra knowledge of label distribution to facilitate the model to focus more on the prediction of minor classes. The motivation of effectively utilizing all training samples to deal with imbalanced FER and increase the performance of minor classes led to the title of our paper: leave no stone unturned.

Table 1: Comparison with different methods on RAF-DB using pre-trained ResNet-18 as backbone. * denotes that we copied the accuracy from the original paper. We arrange the expression classes according to their sample numbers and observe that they exhibit varying levels of difficulty. For instance, despite having a small number of training samples, all methods achieve relatively high performance on the anger class. On the other hand, the disgust and fear classes prove to be the most challenging, and our method achieves the highest accuracy on these two classes. We report the last epoch accuracy of all methods, we also display the best accuracy of our method for reference.

| Method | Conference | Happiness | Neutral | Sadness | Surprise | Disgust | Anger | Fear | Overall | Mean |
|---|---|---|---|---|---|---|---|---|---|---|
| Baseline | - | 95.44 | 88.53 | 85.56 | 83.59 | 58.75 | 78.40 | 59.46 | 87.42 | 78.53 |
| CB [5] | CVPR'19 | 95.11 | 90.74 | 84.73 | 86.93 | 64.38 | 73.46 | 59.46 | 88.04 | 79.26 |
| SCN [42] | CVPR'20 | 94.77 | 90.29 | 80.33 | 86.93 | 60.00 | 76.54 | 45.95 | 86.73 | 76.40 |
| BBN [52] | CVPR'20 | 93.59 | 91.62 | 84.94 | 84.80 | 61.88 | 77.78 | 52.70 | 87.39 | 78.19 |
| PT* [12] | TAFFC'21 | 96.00 | 92.00 | 87.00 | 87.00 | 55.00 | 81.00 | 54.00 | 88.80 | 78.86 |
| RUL [49] | NeurIPS'21 | 95.78 | 87.06 | 86.19 | 89.36 | 65.00 | 83.33 | 64.86 | 88.66 | 81.66 |
| EAC [50] | ECCV'22 | 95.95 | 92.06 | 87.03 | 88.15 | 66.88 | 85.80 | 58.11 | 89.90 | 82.00 |
| Ours | NeurIPS'23 | 96.37 | 89.56 | 89.33 | 87.84 | **66.89** | 80.86 | **66.22** | **89.77** | **82.44** |
| Ours (best) | NeurIPS'23 | 96.03 | 87.79 | 89.33 | 87.23 | 73.13 | 84.57 | 70.27 | 89.80 | 84.05 |

## 4 Experiments

### 4.1 Datasets

RAF-DB [26] has 30,000 facial expression images, which are annotated with basic or compound expressions by 40 trained annotators. In our experiment, we only utilize the 7 basic expressions and these include 12,271 images for training and 3,068 images for testing. We report the mean accuracy on all expressions to evaluate the imbalanced learning performance of different methods.

FERPlus [1] is extended from FER2013 [6] with finer labels. It is collected by the Google search engine with ten crowd-sourced annotators labeling each image. The most voting category is used as the annotation for a fair comparison with other FER methods [43, 37]. We utilize the same 7 classes with RAF-DB, which results in 24,941 images for training and 3,137 images for testing.

AffectNet [32] is a large-scale FER dataset, which contains eight expressions (7 basic expressions and contempt). There are a total of 286,564 training images and 4,000 test images. We carry out experiments with both 7 basic classes and 8 classes. As the test set of AffectNet is balanced, the mean accuracy on all expressions is the same with the overall accuracy.

### 4.2 Implementation details

To make fair comparisons with other methods, we compare all methods under the same backbone of MS-Celeb-1M [8] pre-trained ResNet-18. We also test the effectiveness of our method under different backbones including MobileNet [10], ResNet-50 [9] and Swin Transformer [28]. The facial expression images are detected and aligned using MTCNN [47]. The image size is $224 \times 224$, the learning rate is set to $0.0001$. We use Adam [17] optimizer with weight decay of $0.0001$ and ExponentialLR learning rate scheduler with a gamma of $0.9$. The max training epoch $T_{max}$ is set to 60. The consistency weight $\lambda$ is set to 2 and the smooth parameter $\alpha$ is set to 0.1 according to the ablation study in section 4.7. All experiments are conducted on 4 NVIDIA RTX 2080Ti.

### 4.3 Experiments on imbalanced FER datasets

We conduct experiments on the RAF-DB dataset to evaluate the performance of different methods in imbalanced learning. We report the classification accuracy of each class and the overall and mean accuracy to assess their effectiveness. The baseline method involves training with the pre-trained ResNet-18 without additional modules. We compare our method with imbalanced learning methods CB, BBN, and state-of-the-art FER methods SCN, PT, RUL, and EAC. To obtain the mean accuracy, we re-implement these methods based on their open-source code. The results in Table 1 demonstrate that our method achieves the highest overall accuracy of $89.77\%$ and mean accuracy of $82.44\%$ on the RAF-DB dataset using ResNet-18 as the backbone.

Table 2: Comparison with different methods on AffectNet using pre-trained ResNet-18 as backbone. We carry out experiments with both 7 and 8 classes.

| Method | Conference | Happiness | Neutral | Sadness | Anger | Surprise | Fear | Disgust | Contempt | Mean |
|--------|-----------|-----------|---------|---------|-------|----------|------|---------|----------|------|
| SCN [42] | CVPR'20 | 95.20 | 82.70 | 44.20 | 56.30 | 35.80 | 38.00 | 20.90 | - | 53.30 |
| BBN [52] | CVPR'20 | 87.00 | 57.10 | 66.80 | 58.30 | 54.90 | 71.10 | 30.10 | - | 60.76 |
| RUL [49] | NeurIPS'21 | 90.50 | 62.40 | 64.70 | 69.30 | 60.80 | 49.00 | 34.20 | - | 61.56 |
| EAC [50] | ECCV'22 | 91.40 | 64.50 | 65.70 | 66.30 | 61.60 | 60.90 | 45.80 | - | 65.17 |
| Ours | NeurIPS'23 | 86.20 | 59.00 | 64.20 | 66.50 | 57.80 | **64.50** | **61.90** | - | **65.73** |
| SCN [42] | CVPR'20 | 94.60 | 74.90 | 58.20 | 63.80 | 40.90 | 43.20 | 30.80 | 2.20 | 51.08 |
| BBN [52] | CVPR'20 | 78.40 | 58.40 | 60.60 | 67.70 | 59.40 | 55.00 | 37.00 | 46.70 | 57.90 |
| RUL [49] | NeurIPS'21 | 71.00 | 63.40 | 46.60 | 54.90 | 53.70 | 58.60 | 44.70 | 47.70 | 55.08 |
| EAC [50] | ECCV'22 | 84.00 | 58.80 | 65.00 | 65.90 | 62.20 | 60.30 | 46.10 | 41.90 | 60.53 |
| Ours | NeurIPS'23 | 78.60 | 54.30 | 63.80 | 59.50 | 57.60 | **64.10** | **59.40** | **60.00** | **62.16** |

Based on the accuracy of each class, which is not commonly reported in previous works, we observe varying levels of difficulty among the expression classes. For instance, despite having very few training samples, all methods achieve relatively high performance on the anger class. We speculate that the distinct features associated with anger make it easier for FER models to detect. On the other hand, the disgust and fear classes prove to be the most challenging, with all methods exhibiting low accuracy. However, our method outperforms others on these two classes, achieving accuracies of 66.89% and 66.22% respectively. This highlights the effectiveness of our approach in extracting extra knowledge from both major and minor-class samples to address the imbalanced learning problem.

We further carry out experiments on AffectNet, which is one of the largest and most imbalanced datasets available for FER. From the results in Table 2, we could draw the conclusion that our method achieves the best mean accuracy on the test set under both 7 or 8 classes. Besides, we notice that our method improves existing methods on the minor classes of fear (Fea), disgust (Dis), contempt (Con) by remarkable margins, which illustrates that our method is more suitable for the imbalanced learning of FER task. Our method even achieves 60.00% accuracy on the contempt class under 8 classes. Furthermore, our method clearly decreases the test accuracy gap between happy (78.60%) and contempt (60.00%) compared with other methods under 8 classes, which means our method is fairer and achieves a more balanced test accuracy.

### 4.4 Experiments with different imbalance factors

Following existing imbalanced learning methods [5, 33, 2] in the image classification field, we also construct imbalanced FER datasets with different imbalance factors. The definition of the imbalance factor of a dataset is following [5] as the number of training samples in the largest class divided by the smallest. Given the imbalance factor, the imbalanced FER datasets are created by reducing the number of training samples per class according to an exponential function $n = n_l \mu^l$, where $l$ is the class index, $n_l$ is the original number of training images and $\mu \in (0, 1)$. Due to space limitation, the sample number of each class is summarized in the supplementary material. We evaluate our method on RAF-DB and FERPlus datasets with imbalance factors ranging from 50 to 150, as shown in Table 3 and Table 4. The results demonstrate the superior performance of our method across different imbalance factors. Specifically, compared to the state-of-the-art FER method EAC, our method consistently improves upon it with 4.09%, 3.26%, 1.67% and 2.13%, 2.62%, 4.15% regrading mean accuracy of all classes on RAF-DB and FERPlus respectively.

### 4.5 Different backbones

We evaluate the generalization ability of our method by combining it with four different backbones: MobileNet, ResNet-18, ResNet-50, and Tiny Swin Transformer, on RAF-DB. The results consistently demonstrate the improvement in imbalanced learning performance across different backbones. We also observe that different backbones have a notable effect on the performance of imbalanced learning. Notably, when combined with Tiny Swin Transformer, our method achieves state-of-the-art performance with accuracy of 71.62% and 85.00% on the most difficult expression classes of fear and disgust, respectively, as well as an overall accuracy of 92.31% and a mean accuracy of 87.71%.

Table 3: Comparison with other methods on RAF-DB with different imbalance factors. Disgust and fear are the most difficult classes. Our method achieves the highest accuracy on the overall, mean accuracy and the accuracy on the most difficult classes under different imbalance factors.

| Method | Imbalance | Happiness | Neutral | Sadness | Surprise | Disgust | Anger | Fear | Overall | Mean |
|---|---|---|---|---|---|---|---|---|---|---|
| Baseline | 50 | 95.95 | 87.35 | 79.08 | 84.19 | 39.38 | 64.20 | 2.70 | 83.28 | 64.69 |
| BBN | 50 | 93.59 | 91.91 | 81.80 | 82.98 | 41.25 | 71.60 | 37.84 | 85.01 | 71.57 |
| EAC | 50 | 95.53 | 93.82 | 82.01 | 89.06 | 50.00 | 70.99 | 29.73 | 87.09 | 73.02 |
| Ours | 50 | 96.37 | 90.00 | 85.36 | 85.41 | **53.75** | 73.46 | **55.41** | **87.65** | **77.11** |
| Baseline | 100 | 97.72 | 87.94 | 73.85 | 81.76 | 10.63 | 54.94 | 0.00 | 80.96 | 58.12 |
| BBN | 100 | 94.94 | 93.38 | 71.34 | 82.37 | 36.88 | 65.43 | 31.08 | 83.44 | 67.92 |
| EAC | 100 | 95.27 | 92.06 | 83.68 | 89.97 | 36.88 | 62.35 | 28.38 | 85.79 | 69.80 |
| Ours | 100 | 96.37 | 91.18 | 82.85 | 86.63 | **44.38** | 65.43 | **44.59** | **86.47** | **73.06** |
| Baseline | 150 | 95.86 | 90.29 | 75.73 | 77.51 | 9.38 | 46.91 | 0.00 | 80.11 | 56.53 |
| BBN | 150 | 94.85 | 93.53 | 74.69 | 81.46 | 30.00 | 55.56 | 28.38 | 82.92 | 65.49 |
| EAC | 150 | 96.20 | 91.62 | 77.82 | 79.64 | 36.25 | 59.88 | 39.19 | 84.13 | 68.66 |
| Ours | 150 | 96.62 | 91.91 | 79.29 | 83.89 | **36.25** | 61.11 | **43.24** | **85.20** | **70.33** |

Table 4: Comparison with other methods on FERPlus with different imbalance factors. Our method achieves the highest accuracy on the overall, mean accuracy and the accuracy on the most difficult classes (fear and disgust) under different imbalance factors.

| Method | Imbalance | Neutral | Happiness | Surprise | Sadness | Anger | Fear | Disgust | Overall | Mean |
|---|---|---|---|---|---|---|---|---|---|---|
| Baseline | 50 | 86.42 | 93.17 | 89.14 | 76.56 | 83.88 | 46.99 | 22.22 | 85.85 | 71.20 |
| BBN | 50 | 84.31 | 91.38 | 93.18 | 77.60 | 84.98 | 54.22 | 33.33 | 85.59 | 74.14 |
| EAC | 50 | 90.09 | 95.63 | 90.15 | 76.30 | 84.62 | 49.40 | 33.33 | 88.11 | 74.22 |
| Ours | 50 | 91.19 | 94.06 | 91.67 | 79.95 | 82.05 | **56.63** | **38.89** | **88.68** | **76.35** |
| Baseline | 100 | 90.32 | 93.56 | 87.63 | 66.80 | 78.21 | 46.99 | 22.22 | 85.43 | 69.39 |
| BBN | 100 | 88.62 | 91.71 | 93.18 | 74.74 | 81.32 | 51.81 | 38.89 | 86.48 | 74.32 |
| EAC | 100 | 90.73 | 95.30 | 90.66 | 74.48 | 81.32 | 53.45 | 27.78 | 87.87 | 73.39 |
| Ours | 100 | 91.56 | 94.85 | 92.42 | 77.34 | 82.78 | **54.22** | **38.89** | **88.81** | **76.01** |
| Baseline | 150 | 91.28 | 93.84 | 89.14 | 63.02 | 78.75 | 44.58 | 11.11 | 85.49 | 67.39 |
| BBN | 150 | 90.09 | 92.61 | 93.43 | 67.19 | 79.12 | 46.99 | 33.33 | 86.01 | 71.82 |
| EAC | 150 | 93.12 | 94.74 | 89.65 | 71.35 | 78.75 | 39.76 | 22.22 | 87.41 | 69.94 |
| Ours | 150 | 93.94 | 94.51 | 90.40 | 71.88 | 79.12 | **55.42** | **33.33** | **88.30** | **74.09** |

Table 5: The performance of our method under different backbones. We find that backbones have a significant influence on the accuracy. Our method consistently improves the performance under different backbones and achieves the state-of-the-art overall accuracy of $92.31\%$ and mean accuracy of $87.71\%$ with Tiny Swin Transformer (Swin-T).

| Backbone | Happiness | Neutral | Sadness | Surprise | Disgust | Anger | Fear | Overall | Mean |
|---|---|---|---|---|---|---|---|---|---|
| MobileNet | 93.84 | 83.09 | 77.62 | 88.75 | 45.00 | 75.31 | 56.76 | 83.96 | 74.34 |
| MobileNet+Ours | 94.26 | 88.82 | 81.38 | 83.28 | **59.38** | 74.07 | **59.46** | **86.15** | **77.24** |
| ResNet-18 | 95.44 | 88.53 | 85.56 | 83.59 | 58.75 | 78.40 | 59.46 | 87.42 | 78.53 |
| ResNet-18+Ours | 96.37 | 89.56 | 89.33 | 87.84 | **66.89** | 80.86 | **66.22** | **89.77** | **82.44** |
| ResNet-50 | 94.77 | 87.79 | 87.03 | 85.71 | 68.75 | 84.57 | 60.81 | 88.33 | 81.35 |
| ResNet-50+Ours | 95.95 | 87.65 | 89.75 | 88.75 | **80.63** | 85.19 | **66.22** | **90.29** | **84.88** |
| Swin-T | 97.05 | 91.62 | 87.87 | 90.27 | 78.75 | 86.42 | 60.81 | 91.30 | 84.68 |
| Swin-T+Ours | 96.96 | 92.06 | 88.28 | 92.40 | **85.00** | 87.65 | **71.62** | **92.31** | **87.71** |

Table 6: Ablation study of our proposed two modules re-balanced attention consistency (RAC) and re-balanced smooth labels (RSL). Both of the two modules can improve the performance based on the baseline, while they can cooperate to achieve the best performance.

| RAC | RSL | Happiness | Neutral | Sadness | Surprise | Disgust | Anger | Fear | Overall | Mean |
|---|---|---|---|---|---|---|---|---|---|---|
| | | 95.44 | 88.53 | 85.56 | 83.59 | 58.75 | 78.40 | 59.46 | 87.42 | 78.53 |
| ✓ | | 96.29 | 89.26 | 87.87 | 88.75 | 65.63 | 76.54 | 59.46 | 89.08 | 80.54 |
| | ✓ | 94.60 | 90.44 | 85.15 | 82.07 | 57.50 | 80.86 | 64.86 | 87.48 | 79.35 |
| ✓ | ✓ | 96.37 | 89.56 | 89.33 | 87.84 | **66.89** | 80.86 | **66.22** | **89.77** | **82.44** |

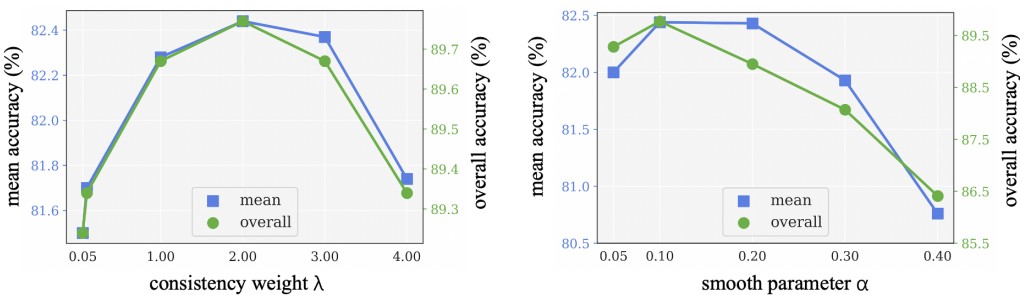

Figure 3: The hyperparameter study of the consistency weight $\lambda$ and the smooth parameter $\alpha$.

## 4.6 Ablation study

We conduct an ablation study on RAF-DB to evaluate the contribution of each proposed module. The results in Table 6 demonstrate the effectiveness of both the re-balanced attention consistency (RAC) and re-balanced smooth labels (RSL) in improving the baseline method's performance. Interestingly, we observe that utilizing only the RAC module achieves superior performance compared to using only the RSL module. This could be attributed to the additional information provided by the re-balanced attention map consistency. Moreover, combining both RAC and RSL modules results in even better performance, indicating their effective collaboration in addressing the imbalanced FER task.

## 4.7 Hyperparameter study

We carry out experiments on RAF-DB to study the effect of different hyperparameters to our method. We plot the results in Figure 3.

**Consistency weight** $\lambda$ The results demonstrate that our method exhibits low sensitivity to the consistency weight $\lambda$, with both the mean accuracy and overall accuracy varying within a small range of $1\%$ as $\lambda$ changes from $0.05$ to $4$. Notably, the optimal value for $\lambda$ in our method is found to be $2$, indicating that larger values may excessively prioritize consistency loss over classification loss, potentially leading to a decrease in classification accuracy. Conversely, smaller values of $\lambda$ may fail to effectively regulate the model in extracting additional knowledge related to minor classes from all samples, thus negatively impacting accuracy.

**Smooth parameter** $\alpha$ The smooth parameter, ranging from $0$ to $1$, determines the strength of the latent truth (set as $1 - \alpha$). We evaluate different values of $\alpha$ from $0.05$ to $0.4$ and find that the optimal value is $\alpha = 0.1$. A larger $\alpha$ negatively impacts performance as the excessive smooth effect hampers the model's ability to learn useful information. On the other hand, a smaller $\alpha$ fails to effectively utilize the prior knowledge of label distribution to prioritize minor classes.

## 4.8 Visualization results

We provide visualization results to illustrate the effectiveness of our proposed method. The learned attention maps, as shown in Figure 4, reveal three key observations. First, our method consistently learns attention maps that are more consistent across different transformations, enabling the FER model to capture transformation invariant information for various expression classes. Second, our method effectively extracts additional knowledge related to minor classes (e.g., disgust and fear) from samples of major classes (e.g., sadness and surprise), as there are shared features between major

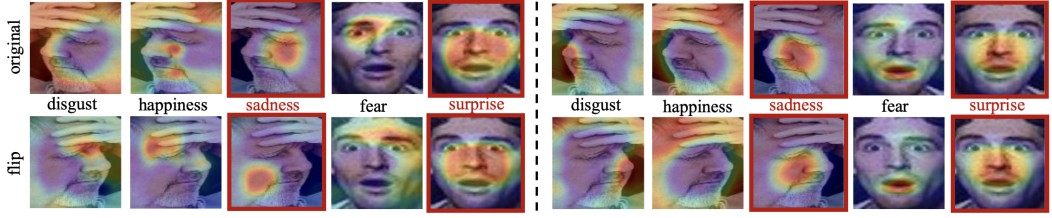

Figure 4: The attention maps corresponding to different classes learned by different methods. We utilize the attention map of a certain class to mine transformation invariant information of that class. The attention maps of the labels are marked by red. Attention maps learned by our method are more consistent before and after the flip transformation across all different classes. Furthermore, shown in the last two columns, our method can mine extra knowledge related to the minor classes like fear (the open mouth feature) from the samples of major classes of surprise.

Table 7: Other transformation methods for re-balanced attention consistency.

| Method | Happiness | Neutral | Sadness | Surprise | Disgust | Anger | Fear | Overall | Mean |
|---|---|---|---|---|---|---|---|---|---|
| Intensity | 95.02 | 86.47 | 82.01 | 82.98 | 63.13 | 72.84 | 56.76 | 86.05 | 77.03 |
| Scaling | 95.78 | **91.91** | 84.31 | 85.41 | **75.00** | **82.10** | 60.81 | 89.37 | 82.19 |
| Ours | **96.37** | 89.56 | **89.33** | **87.84** | 66.89 | 80.86 | **66.22** | **89.77** | **82.44** |

and minor classes in FER. For instance, the first column in the right part of Figure 4 demonstrates how our method captures the mouth corner feature associated with disgust from a sample labeled as sadness. Furthermore, the last two columns in the figure show that our method identifies the open mouth feature shared by fear and surprise, allowing us to extract fear-related features from surprise samples. More results in the Supp. material. Third, our method produces non-overlapping attention maps for different classes, in contrast to the baseline method. For example, for the sample labeled as sadness, the attention maps for happiness and sadness learned by our method do not overlap, while they overlap in the baseline method. This indicates that the attention maps learned by our method are more meaningful and distinct.

## 4.9   Other transformations

Flipping of the images is shown to introduce the notion of re-balanced attention consistency. In this section, we investigate whether some other transformations (e.g., scaling, intensity attenuation, or gain) work well under the imbalanced FER task. The results on RAF-DB in Table 7 illustrate that intensity transformation performs poorly, while scaling performs well, which almost surpasses our method. The reason lies in that attention map consistency regularizes the model to focus on the same regions before and after the transformation, which incorporates spatial information as the attention map in our method has height and width dimensions. Thus, the transformation should be spatial-related transformation to maximize the function of the method.

## 5   Conclusion

In this paper, we investigate the imbalanced learning problem in facial expression recognition (FER). We observe that existing imbalanced learning methods tend to improve performance on minor classes at the expense of major classes. Motivated by the label distribution learning characteristic of FER, we propose a novel approach to extract additional knowledge about minor classes from both major and minor class samples. This allows us to enhance the performance on minor classes while maintaining high performance on major classes. Our method consists of two modules: re-balanced attention consistency and re-balanced smooth labels, which regulate attention maps and classification loss, respectively. Instead of relying on traditional over-sampling or under-sampling techniques, our method effectively utilizes all training samples and incorporates prior knowledge of the imbalanced data distribution to prioritize minor classes. Through extensive experiments on various imbalanced FER datasets and with different backbones, we validate the effectiveness of our proposed method.

## Acknowledgments and Disclosure of Funding

We sincerely thank all the reviewers who have given us lots of valuable suggestions for the improvement of our paper. This work was supported in part by the BUPT Excellent Ph.D. Students Foundation No.CX2023111 and in part by scholarships from China Scholarship Council (CSC) under Grant CSC No.202206470048.

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
