# Leave No Stone Unturned: Mine Extra Knowledge for Imbalanced Facial Expression Recognition Supplementary Material

## 1 Additional results of other methods on imbalanced FER datasets

We present the results of several methods, namely CB [1], SCN [3], RUL [4], and SOFT [2], on imbalanced RAF-DB and FERPlus datasets. The results are summarized in Table 1 and Table 2 for RAF-DB and FERPlus, respectively. Based on the results from the two FER datasets, we have drawn some conclusions. Firstly, it is evident that different expression classes possess varying levels of difficulty. When arranging the expression classes in descending order according to their training samples, if they had similar difficulty levels, the test accuracy should also follow a descending trend. However, both Table 1 and Table 2 demonstrate that different expression classes exhibit distinct levels of difficulty. For instance, in Table 1, all the FER methods achieve higher performance on the surprise class compared to the sadness class, despite the sadness class having more training samples. Similarly, in Table 2, FER methods achieve higher performance on the happiness class than on the neutral class, despite the neutral class having more training samples. These results lead us to conclude that different expression classes indeed present varying levels of difficulty. Generally, expression classes with distinct features, such as surprise and anger, tend to be easier compared to sadness, even when they have similar training samples. Additionally, the happiness class is easier than the neutral class when they possess similar training samples, likely due to the greater ambiguity of the neutral class in its semantic meaning. Lastly, the fear and disgust classes emerge as the most challenging expression classes, as they have the fewest training samples and lack clearly defined features in comparison to other expression classes.

The results presented in Table 1 and Table 2 also demonstrate the superior performance of our method in terms of both mean accuracy and overall accuracy on the test set. Additionally, our method achieves exceptional performance on the two most challenging expression classes, fear and disgust. It is worth noting that SOFT and SCN exhibit relatively lower mean accuracy across different imbalanced datasets. This can be attributed to the fact that SOFT and SCN modify the training labels during the training process, based on the performance of the training model. In the case of imbalanced training sets, where the model struggles with the minority classes, the labels are more likely to be adjusted towards the majority classes. Consequently, SOFT and SCN achieve high overall accuracy but suffer in terms of mean accuracy. This observation emphasizes the importance of not solely relying on overall accuracy as a comprehensive evaluation metric for different FER methods. Furthermore, we observe that CB and RUL perform well in terms of mean accuracy. CB addresses the imbalanced learning problem by utilizing imbalanced weights to balance the cross-entropy loss. This approach effectively mitigates the impact of imbalanced training data. On the other hand, RUL incorporates uncertainty-weighted feature mixup, which involves mixing samples from different classes during training. This strategy enhances the model's ability to extract information from the minority classes, leading to improved mean accuracy. In summary, our method demonstrates superior performance across various evaluation metrics, including mean accuracy and overall accuracy, outperforming alternative approaches such as SOFT, SCN, CB, and RUL.

Table 1: Comparison with other methods on RAF-DB with different imbalance factors. Our method achieves the highest accuracy on the overall and the mean accuracy under different imbalance factors.

| Method | Imbalance | Happiness | Neutral | Sadness | Surprise | Disgust | Anger | Fear | Overall | Mean |
|---|---|---|---|---|---|---|---|---|---|---|
| SOFT | 50 | 94.09 | 90.44 | 59.62 | 82.37 | 12.50 | 58.02 | 0.00 | 78.23 | 56.72 |
| SCN | 50 | 96.03 | 89.12 | 78.87 | 86.02 | 36.25 | 63.58 | 0.00 | 83.60 | 64.27 |
| CB | 50 | 94.94 | 89.85 | 82.01 | 87.84 | 51.88 | 75.31 | 41.89 | 86.47 | 74.82 |
| RUL | 50 | 94.43 | 91.91 | 78.45 | 88.15 | 50.63 | 67.28 | 28.38 | 85.40 | 71.32 |
| Ours | 50 | 96.37 | 90.00 | 85.36 | 85.41 | **53.75** | 73.46 | **55.41** | **87.65** | **77.11** |
| SOFT | 100 | 95.02 | 89.26 | 52.30 | 80.55 | 2.50 | 42.59 | 0.00 | 75.65 | 51.75 |
| SCN | 100 | 95.86 | 92.65 | 76.36 | 84.19 | 11.25 | 56.79 | 0.00 | 82.07 | 59.59 |
| CB | 100 | 96.71 | 90.74 | 71.76 | 78.42 | 32.50 | 64.20 | 47.30 | 83.28 | 68.80 |
| RUL | 100 | 96.46 | 85.44 | 80.75 | 85.11 | 39.38 | 59.26 | 39.19 | 84.03 | 69.37 |
| Ours | 100 | 96.37 | 91.18 | 82.85 | 86.63 | **44.38** | 65.43 | **44.59** | **86.47** | **73.06** |
| SOFT | 150 | 96.37 | 87.21 | 63.60 | 77.51 | 1.25 | 28.40 | 0.00 | 76.34 | 50.62 |
| SCN | 150 | 96.62 | 93.09 | 72.38 | 75.08 | 0.00 | 49.38 | 0.00 | 79.89 | 55.22 |
| CB | 150 | 97.72 | 81.62 | 77.41 | 83.28 | **37.50** | 61.11 | 39.19 | 82.95 | 68.26 |
| RUL | 150 | 96.71 | 86.32 | 79.92 | 84.19 | 33.75 | 64.20 | 9.46 | 83.34 | 64.94 |
| Ours | 150 | 96.62 | 91.91 | 79.29 | 83.89 | 36.25 | 61.11 | **43.24** | **85.20** | **70.33** |

Table 2: Comparison with other methods on FERPlus with different imbalance factors. Our method achieves the highest accuracy on the overall, mean accuracy and the accuracy on the hardest classes (disgust and fear) under different imbalance factors.

| Method | Imbalance | Neutral | Happiness | Surprise | Sadness | Anger | Fear | Disgust | Overall | Mean |
|---|---|---|---|---|---|---|---|---|---|---|
| SOFT | 50 | 90.28 | 92.72 | 92.42 | 73.18 | 86.08 | 32.53 | 0.00 | 86.74 | 66.74 |
| SCN | 50 | 87.27 | 93.17 | 92.17 | 73.44 | 82.05 | 40.96 | 0.00 | 85.85 | 67.01 |
| CB | 50 | 84.40 | 94.51 | 90.66 | 79.17 | 84.62 | 55.42 | 33.33 | 86.39 | 74.59 |
| RUL | 50 | 87.76 | 94.18 | 91.41 | 78.91 | 85.71 | 56.63 | 33.33 | 87.79 | 75.42 |
| Ours | 50 | 91.19 | 94.06 | 91.67 | 79.95 | 82.05 | **56.63** | **38.89** | **88.68** | **76.35** |
| SOFT | 100 | 93.94 | 95.52 | 86.62 | 62.76 | 78.02 | 30.12 | 0.00 | 86.04 | 63.85 |
| SCN | 100 | 90.60 | 91.71 | 91.16 | 69.53 | 81.32 | 45.78 | 0.00 | 86.07 | 67.16 |
| CB | 100 | 91.10 | 93.62 | 88.13 | 69.27 | 79.85 | 55.42 | 38.89 | 86.55 | 73.75 |
| RUL | 100 | 91.38 | 95.41 | 89.39 | 70.31 | 82.78 | 54.22 | 38.89 | 87.70 | 74.63 |
| Ours | 100 | 91.56 | 94.85 | 92.42 | 77.34 | 82.78 | **54.22** | **38.89** | **88.81** | **76.01** |
| SOFT | 150 | 94.95 | 95.63 | 88.13 | 54.95 | 76.19 | 30.12 | 0.00 | 85.50 | 62.85 |
| SCN | 150 | 92.56 | 94.51 | 89.65 | 54.69 | 74.73 | 30.12 | 0.00 | 84.48 | 62.32 |
| CB | 150 | 92.57 | 93.84 | 86.62 | 63.28 | 78.02 | 46.99 | 27.78 | 85.75 | 69.87 |
| RUL | 150 | 92.75 | 94.96 | 89.90 | 67.97 | 80.22 | 54.22 | 33.33 | 87.63 | 73.34 |
| Ours | 150 | 93.94 | 94.51 | 90.40 | 71.88 | 79.12 | **55.42** | 33.33 | **88.30** | **74.09** |

Table 3: Comparison with different methods on FERPlus using pre-trained ResNet-18 as backbone. Our method achieves the best overall accuracy and mean accuracy.

| Method | Conference | Neutral | Happiness | Surprise | Sadness | Anger | Fear | Disgust | Overall | Mean |
|---|---|---|---|---|---|---|---|---|---|---|
| Baseline | - | 89.36 | 94.06 | 88.38 | 68.23 | 80.22 | 50.60 | 44.44 | 85.91 | 73.61 |
| CB [1] | CVPR'19 | 91.56 | 93.06 | 87.12 | 74.74 | 83.88 | 53.01 | 44.44 | 87.41 | 75.40 |
| SCN [3] | CVPR'20 | 90.70 | 94.18 | 88.64 | 63.80 | 83.52 | 42.17 | 0.00 | 85.81 | 66.14 |
| BBN [6] | CVPR'20 | 88.26 | 94.18 | 92.68 | 75.26 | 83.15 | 53.01 | 38.89 | 87.25 | 75.06 |
| RUL [4] | NeurIPS'21 | 89.52 | 94.85 | 89.90 | 77.86 | 85.71 | 56.63 | 33.33 | 88.30 | 75.40 |
| SOFT [2] | ECCV'22 | 92.94 | 94.40 | 90.15 | 67.71 | 83.15 | 38.55 | 0.00 | 87.09 | 66.72 |
| EAC [5] | ECCV'22 | 88.36 | 94.62 | 90.91 | 78.91 | 82.78 | 51.81 | 33.33 | 88.36 | 74.73 |
| Ours | - | 92.84 | 94.51 | 91.16 | 77.08 | 82.42 | **56.63** | **44.44** | **89.03** | **77.01** |

Table 4: The distribution of the imbalanced RAF-DB.

| Imbalance | Happiness | Neutral | Sadness | Surprise | Disgust | Anger | Fear | Total |
|---|---|---|---|---|---|---|---|---|
| - | 4772 | 2524 | 1982 | 1290 | 717 | 705 | 281 | 12271 |
| 50 | 4772 | 2108 | 1382 | 751 | 349 | 286 | 95 | 9743 |
| 100 | 4772 | 1878 | 1097 | 531 | 220 | 161 | 48 | 8707 |
| 150 | 4772 | 1755 | 959 | 434 | 168 | 115 | 32 | 8235 |

Table 5: The distribution of the imbalanced FERPlus.

| Imbalance | Neutral | Happiness | Surprise | Sadness | Anger | Fear | Disgust | Total |
|---|---|---|---|---|---|---|---|---|
| - | 8740 | 7287 | 3149 | 3014 | 2100 | 532 | 119 | 24941 |
| 50 | 5950 | 5950 | 3149 | 3014 | 2100 | 532 | 119 | 20814 |
| 100 | 8740 | 6923 | 2842 | 2584 | 1710 | 412 | 87 | 23280 |
| 150 | 8740 | 6465 | 2482 | 2109 | 1305 | 293 | 58 | 21452 |

We further provide a comparison of results on the original FERPlus dataset, as displayed in Table 3. By employing ResNet-18 as the backbone, our method achieves the highest accuracy among all other methods, particularly excelling in the minor classes of fear and disgust. Additionally, our method attains the best overall accuracy of $89.03\%$ and simultaneously achieves the highest mean accuracy of $77.01\%$.

## 2 The distribution of the imbalanced FER datasets

We present the distribution of training samples across different expression classes in RAF-DB and FERPlus, as shown in Table 4 and Table 5. To create imbalanced FER datasets, we apply an exponential function $n = n_l \mu^l$ to reduce the number of training samples per class, where $l$ represents the class index, $n_l$ denotes the original number of training images for class $l$, and $\mu \in (0, 1)$. It is important to note that since the original FERPlus dataset already exhibits a significant imbalance factor exceeding 50, we instead reduce the number of training samples in the major classes (neutral and happiness) to construct a more balanced FER dataset.

## 3 Additional visualization results

We show more results of the learned attention maps by the baseline and our method from Figure 1 to Figure 5 to make comparisons. From the results, we mainly draw three conclusions. First, our method learns more transformation-consistent attention maps, as the attention maps on the image before and after flip are focusing on the same area, which means our method can capture more meaningful transformation-invariant information about different expression features. Second, our method can extra knowledge related to minor classes of fear and disgust from other major-class samples to improve the performance of minor classes while not degrading the high performance of major classes. For example, our method extracts the open mouth which is related to the minor-class fear from a samples with the labels of happiness and surprise in Figure 1 and Figure 5, respectively. In Figure 2 and Figure 3, our method extracts the feature of the mouth corner which is related to the minor-class disgust from the samples from major-class sadness. Thirdly, the learned feature maps of different classes exhibit less overlap with each other. In instances where the given image contains no specific information related to a certain class, our learned attention maps are primarily distributed around the periphery of the face. This ensures that the attention maps of the latent truth are not affected, thereby preserving the high performance on major classes. In contrast, the attention maps learned by the baseline method can be distributed anywhere, indicating less meaningful and potentially noisy attention patterns.

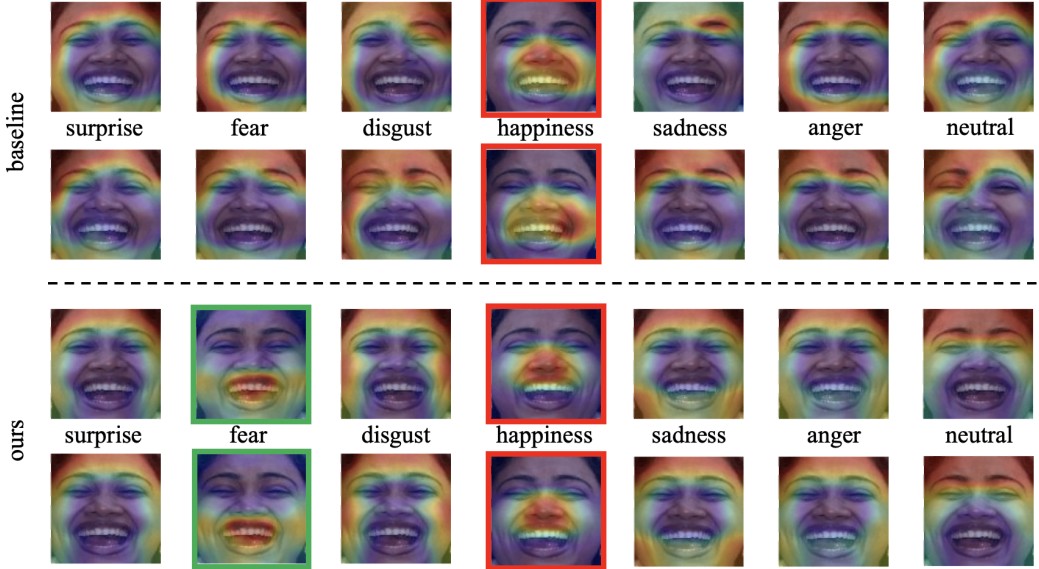

Figure 1: The learned attention maps of different expression classes. The label is marked by red. Our method learns consistent attention maps before and after flip. Furthermore, our method can extract extra knowledge (the feature of open mouth) related to the minor-class fear (marked by green) from the sample of major-class happiness.

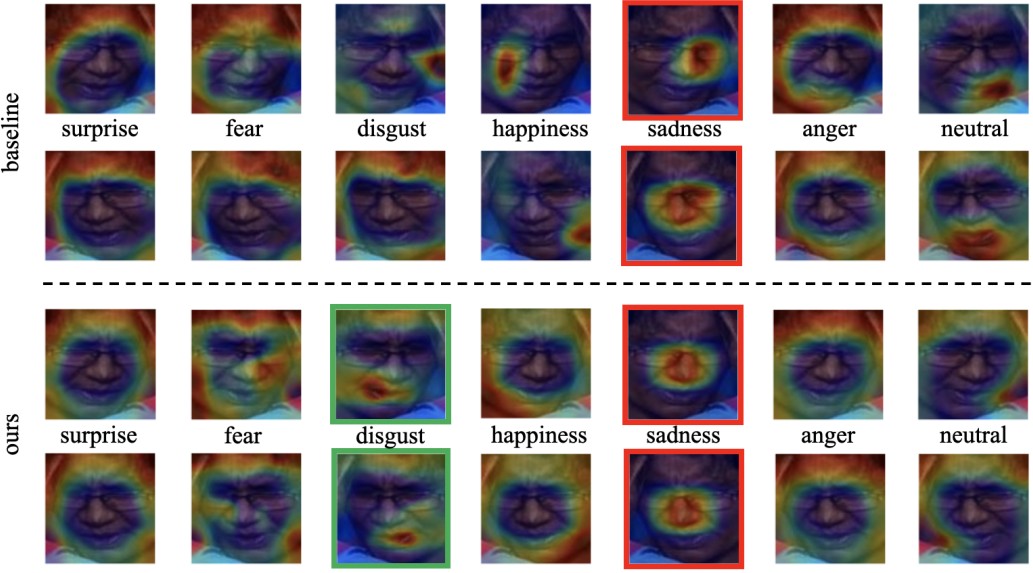

Figure 2: The learned attention maps of different expression classes. The label is marked by red. Our method learns consistent attention maps before and after flip. Furthermore, our method can extract extra knowledge (the feature of the mouth corner) related to the minor-class disgust (marked by green) from the sample of major-class sadness.

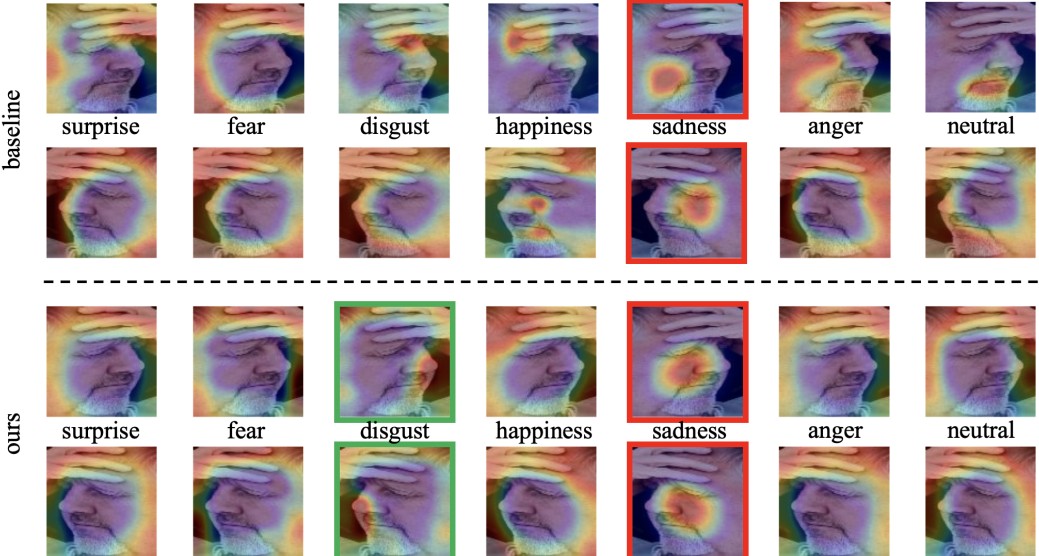

Figure 3: The learned attention maps before and after the flip transformation. The label is marked by red. Our method learns consistent attention maps before and after flip. Furthermore, our method can extract extra knowledge (the feature of the mouth corner) related to the minor-class disgust (marked by green) from the sample of major-class sadness.

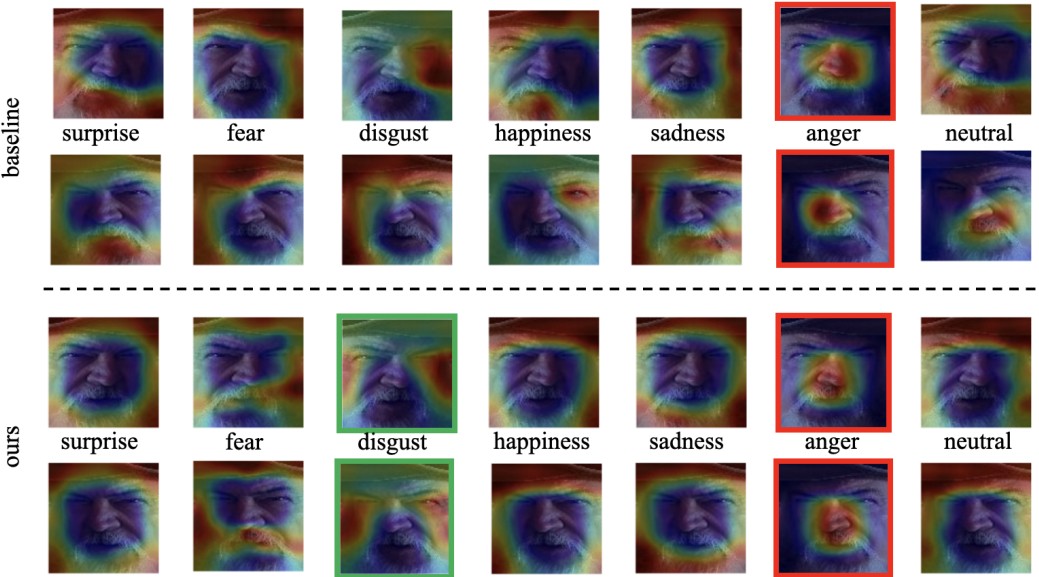

Figure 4: The learned attention maps of different expression classes. The label is marked by red. Our method learns consistent attention maps before and after flip. Furthermore, our method can extract extra knowledge (the feature of raised corners of the eyes) related to the minor-class disgust (marked by green) from the sample of major-class anger.

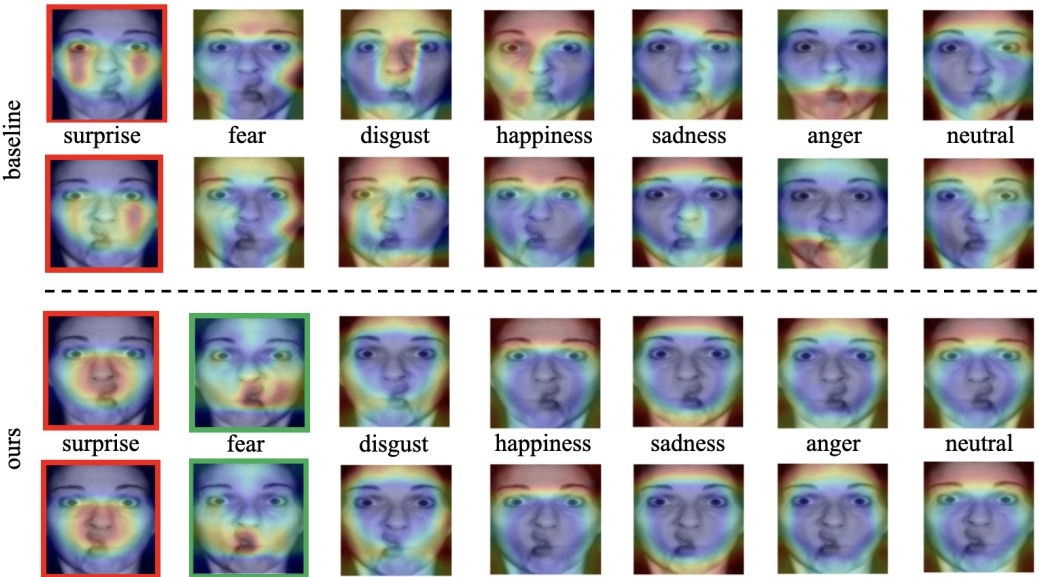

Figure 5: The learned attention maps of different expression classes. The label is marked by red. Our method learns consistent attention maps before and after flip. Furthermore, our method can extract extra knowledge (the feature of open mouth) related to the minor-class fear (marked by green) from the sample of major-class surprise.