# OpenReview forum: "Leave No Stone Unturned: Mine Extra Knowledge for Imbalanced Facial Expression Recognition"
_NeurIPS.cc/2023/Conference — NeurIPS 2023 poster_

### Official Review · Reviewer_ss3S · 2023-06-20

**Soundness:** 3 good
**Presentation:** 3 good
**Contribution:** 3 good
**Rating:** 7
**Confidence:** 4

**Summary:**

This paper deals with the topic of facial  expression recognition (FER). In particular, authors point out the problem of imbalanced class data since most FER data sets will have many more neutral or happy face images than images with other facial expressions.  Authors propose an approach to address this problem building on two ideas. First is that there may be information to be learnt about the minor classes even from the samples from the major classes.   Based on this observation, authors propose a novel attention map rebalancing to regularize the model. As the second idea, they  introduce a label smoothing approach that weights the minor classes more than the major classes. Results  on RAF-DB and FERPlus facial expression image data sets show that the proposed approach is able to outperform state of the art FER approaches.

**Strengths:**

The main idea of learning about the minor classes from training samples in both major and minor classes appears to be novel and is one of the strengths of this paper. The idea of weighting the minor classes more heavily or using smoothed labels have been used in other applications before, but perhaps not in FER.

Another strength of this paper is the  set of numerical results provided. Both data sets being used are state of the art data sets and results clearly indicate that the proposed method outperforms competing methods for FER.

Another strength of the paper is that it is relatively well-written although authors should have more clearly explained the difference between "overall accuracy" and "mean accuracy".

**Weaknesses:**

In Fig. 1, flipping of the images is shown to introduce the notion of re-balanced attention consistency. It is not clear if flipping is the only transformation that makes sense for the FER or if some other transformations (e.g., scaling, intensity attenuation or gain) make sense. That needs to be discussed in more detail.

Another weakness is that the proposed method is somewhat similar to the EAC approach although authors do a good job of clarifying the similarities and differences between their approach and EAC.

The main assertion that there is information to be gained about minor classes from samples in major classes seems to mainly backed up by pointing to the similarities in corresponding attention maps (e.g., open mouth regions for fear and surprise categories). This evidence is subjective and not entirely convincing.

**Questions:**

1. What is the difference between mean accuracy and overall accuracy?

2. Line 106: How do we know that the two types of information are "orthogonal"? Perhaps, characterizing them as "complementary" might be more appropriate.

3. Fig. 1 caption: "--- while do not degrade the high accuracy on major classes" should be "--- while not degrading the high accuracy on major classes"

4. Fig. 2: I assume that the bar corresponding to 0.6 probability is for neutral images --- that needs to be clearly indicated in the figure.

5. Eq. (7): Is the tilde on y() here being used to denote the flipped samples or something else?

6. Shouldn't there be bold-face numbers in every column of Table 1 and other tables?

**Limitations:**

There is no discussion of  limitations in this submission.

Authors may want to add some discussion of the  performances of the proposed approaches based on gender, race and/or age considerations.

---

> ### Author Rebuttal · Authors · 2023-08-09
>
> We are very exhilarated to receive your review. We sincerely thank you for your professional review, which gives us many instructions for polishing our paper. Thanks very much.
>
> **Weaknesses:**
>
> **1.** Thanks for your valuable suggestion. We carried out experiments to study the performance of using other transformations. The results are displayed below.
>
> |Method|Hap|Neu|Sad|Sur|Dis|Ang|Fea|Overall Acc|Mean Acc|
> |--|:-:|:-:|:-:|:-:|:-:|:-:|:-:|:-:|:-:|
> |Intensity|95.02|86.47|82.01|82.98|63.13|72.84|56.76|86.05|77.03|
> |Scaling|95.78|**91.91**|84.31|85.41|**75.00**|**82.10**|60.81|89.37|82.19|
> |Ours|**96.37**|89.56|**89.33**|**87.84**|66.89|80.86|**66.22**|**89.77**|**82.44**|
>
> The results illustrate that intensity transformation performs poorly, while scaling performs well, which almost surpasses our method. The reason lies in that attention map consistency regularizes the model to focus on the same regions before and after the transformation, which incorporates spatial information as the attention map in our method has height and width dimensions. Thus, the transformation should be spatial-related transformation to maximize the function of the method.
>
> **2.** Although the technical details of our RAC module are similar to EAC, the motivation and the task are different. EAC aims to prevent the model from memorizing part of the features to handle label noises, while our RAC aims to mine extra information related to minor classes from all training samples to solve imbalanced learning. We also introduce an RSL module to further enhance performance. Experiment results show that our method distinctly outperforms EAC under the imbalanced learning task.
>
> **3.** Thanks for bringing that up. The evidence is somewhat subjective since facial expression recognition can be subjective and context-dependent, unlike other classification tasks with clear-out boundaries like CIFAR-10. For instance, various annotators of facial expression datasets might yield diverse results for the same image. Our assertion is also supported by [1-3], all of which validate that expression is a continuous statement, implying that a sample could contain information from several classes. Notably, the dataset in [3] comprises expressions such as sadly fearful, sadly disgusted, fearfully surprised, and more. This aligns with our research, where we strive to extract extra information related to minor classes like fear and disgust from major classes like surprise and sadness. Moreover, our Supp. material contains additional visualization results to provide a more intuitive comprehension. We conducted experiments to support our claim. During training, we employed attention map consistency solely on the label class, rather than all classes, for each sample. The results are presented below.
>
> |Method|Hap|Neu|Sad|Sur|Dis|Ang|Fea|Overall Acc|Mean Acc|
> |--|:-:|:-:|:-:|:-:|:-:|:-:|:-:|:-:|:-:|
> |One map|95.02|**90.00**|87.66|85.71|57.50|**80.86**|63.51|88.30|80.04|
> |Ours|**96.37**|89.56|**89.33**|**87.84**|**66.89**|**80.86**|**66.22**|**89.77**|**82.44**|
>
> Our method outperforms the comparison group in almost all cases. The results validate our claim that FER resembles label distribution learning, as solely utilizing attention map consistency on the label class fails to capture additional information from other classes within each given sample.
>
> **Questions:**
>
> **1.** Mean accuracy is calculated as the average value of the test accuracy for each class. Overall accuracy is calculated as the test accuracy for the entire test set. Mean accuracy equals overall accuracy when the test set is balanced, meaning each class has the same number of samples. Both RAF-DB and FERPlus have imbalanced test sets, implying the model could easily achieve high overall accuracy by correctly classifying major classes such as happy. This is unsatisfactory as understanding the minor classes like fear and disgust with negative human emotions is equally crucial.
>
> **2.** Thanks for your valuable advice. We use the term "orthogonal supplement" to convey that previous works primarily address the noise in FER datasets. However, label noises are not connected to the varying sample numbers of each class, which causes the imbalanced learning problem. Nevertheless, your advice to use "complementary" is indeed a superior choice, and we have revised our manuscript.
>
> **3.** We appreciate your suggestion. We have corrected the grammar mistake and conducted proofreading to polish our writing.
>
> **4.** Yes, you're right. We have indicated the neutral class in the new version of Fig. 2 and added it to our manuscript.
>
> **5.** The tilde on y() signifies the re-balanced smooth label, distinct from the original label y(). To enhance clarity, we have replaced tilde y() with y_smooth().
>
> **6.** As we mainly focus on the mean accuracy and the accuracy of minor classes, we did not use bold-face numbers in every column. According to your advice, we used bold-face numbers in every column of our tables and marked the mean accuracy and the accuracy of minor classes using a different color.
>
> **Limitations:**
>
> We added the limitation discussion regarding gender, race, and age to our revised manuscript. Societal and cultural factors might lead to distinct expression patterns, which could impact the model's ability to generalize across genders and races. Things are similar regarding the age factor as facial expressions evolve over time due to muscle tone changes and aging effects.
>
> [1] Sixteen facial expressions occur in similar contexts worldwide. In Nature.
>
> [2] Multi-Dimensional, Nuanced and Subjective-Measuring the Perception of Facial Expressions. In CVPR.
>
> [3] Multi-Label Compound Expression Recognition: C-EXPR Database & Network. In CVPR.

---

> > ### Comment · Reviewer_ss3S · 2023-08-13
> > **Thanks for your rebuttal comments to my review.**
> >
> > Thank you  for carefully responding  to my comments and feedback. I have also read the other reviews and your rebuttal comments to those reviews. Overall, I am convinced that this will be a valuable contribution to NeurIPS 2023 and I will stay with my original rating of Accept.

---

> > > ### Author Response · Authors · 2023-08-13
> > > **Thanks for your time and effort.**
> > >
> > > We sincerely thank you for both your prompt review and feedback. Your professional and thoughtful review provides us with many instructions for improving the quality of our paper. We'd also like to express our gratitude for your dedication in thoroughly reading all other reviews and comments. Your responsible and diligent reviewing is an honor for us.
> > >
> > > We're delighted to learn that you continue to support our paper and recommend its acceptance. Thank you once again for your valuable contributions.

---

### Official Review · Reviewer_N9Q5 · 2023-06-25

**Soundness:** 3 good
**Presentation:** 3 good
**Contribution:** 3 good
**Rating:** 6
**Confidence:** 5

**Summary:**

This paper mainly focuses on solving the imbalanced problem in Facial Expression Recognition (FER). The goal of this paper is to enhance the performance on minor classes without compromising the performance on major classes. The contribution is two fold. Re-balanced attention consistency (RAC) module is proposed to mines extra knowledge pertaining to minor classes from both major and minor samples. Re-balanced smooth labels (RSL) module is proposed to regulate the classification loss and promote balanced learning. Extensive experiments on different datasets and backbones validate the effectiveness of the proposed method.

**Strengths:**

(1) This paper adapts attention map consistency to solve the imbalanced learning problem of FER for the first time, which proves that solving imbalanced FER through re-balanced strategy is promising.

(2) The experiment including various backbones, imbalance and attention visualization is sufficient.

**Weaknesses:**

(1) The purpose of re-balanced attention consistency (RAC) is to extract balanced and transformation invariant knowledge of minor classes from all training samples. The word "knowledge" is unclear and should be more specific. Does it refer to high response values for specific regions on the facial feature map or something else?

(2) The authors' explanation about the role of re-balanced smooth labels (RSL) is to improve performance to regulate the classification loss and promote balanced learning. This explanation is too abstract and difficult for readers to understand. Why is it designed as Eq.(8)? What do the two terms of Eq.(8) represent and why fusing these two terms works? I think the author should provide a more detailed explanation.

(3) The proposed re-balanced attention consistency (RAC) is very similar to the attention consistency used in EAC. The difference is that an additional balance weight is used to re-weight the attention maps. I would like to know how much performance has been improved due to the addition of balance weight. I have looked at the ablation study in Table 5 and can only know the performance improvement of the entire RAC module.

(4) The comparison in Table 1 is questionable. The overall accuracy of EAC is 88.01%, which does not match the number of 89.99% in the original EAC paper (See Table 6 in [46]).

(5) In line 283, the authors think non-overlapping attention maps is better than overlapping ones. Are there any references supporting this viewpoint?

Overall I think the paper is tackling the right problem. However, the unclear description of the proposed modules weakens the contribution of the paper.

**Questions:**

Same as the weakness section.

**Limitations:**

The authors did not discuss the limitation in this paper. I suggest that the authors provide the "maximum capability" of the proposed method. In other words, what degree of imbalance will cause the proposed method to fail.

---

> ### Author Rebuttal · Authors · 2023-08-09
>
> Thanks very much for your thorough review, which helps us a lot to improve our paper.
>
> **Weaknesses:**
>
> **1.** Yes, the knowledge pertains to attention regions on the feature map. The transformation invariant knowledge ensures the FER model to focus on the same region before and after the transformation, which improves accuracy according to [7, 46]. The novelty and motivation of our paper are rooted in our idea of extracting useful minor-class features across all training samples, instead of simply introducing attention map consistency for the imbalanced FER task.
>
> **2.** The motivation for RSL is shown in Fig.2. By setting the smooth label inversely proportional to the sample number of each class, we create an imbalanced smooth label with larger smooth labels for minor classes compared to major classes. To achieve minimum loss during training, the FER model will predict more logits into the minor classes. As stated by Reviewer ss3S, this is akin to weighing the minor classes more heavily during training to improve the model's performance on them. The computation of Eq.(8) follows Fig.2 and is derived from Eq.(3) of label smooth in paper [1]. In Eq.(8), $\tilde{y}(i,l)$ is the $l$-th class value of the re-balanced one-hot label of $x_{i}$. The first item represents the original value of the one-hot label with a weight of $(1-\alpha)$. The second item is the smooth label introduced by our method, where $(\alpha/L)$ is the weight for normalization. $B_{l}$ is our re-balanced weight. Eq.(8) does not represent a fusion of two terms; it is the computation of our introduced RSL.
>
> **3.** While the technical implementation of RAC is similar to EAC, the motivation and the resolved task are quite distinct. EAC focuses on preventing the model from remembering part of the features to mitigate label noise, while our RAC aims to mine extra information of minor classes from all samples to promote balanced learning. As we introduce the RSL module in our paper, to solely evaluate the effectiveness of the balance weight, we combine EAC with our RSL module as the comparison group. The results on RAF-DB show that our method outperforms EAC+RSL on both overall and mean accuracy.
>
> |Method|Hap|Neu|Sad|Sur|Dis|Ang|Fea|Overall Acc|Mean Acc|
> |--|:-:|:-:|:-:|:-:|:-:|:-:|:-:|:-:|:-:|
> |EAC+RSL|95.95|88.24|88.91|86.63|65.63|79.63|63.51|88.92|81.21|
> |Ours|**96.37**|**89.56**|**89.33**|**87.84**|**66.89**|**80.86**|**66.22**|**89.77**|**82.44**|
>
> We further show the results when the imbalance factor is 150. The results illustrate that our method also outperforms EAC+RSL in most cases.
>
> |Method|Hap|Neu|Sad|Sur|Dis|Ang|Fea|Overall Acc|Mean Acc|
> |--|:-:|:-:|:-:|:-:|:-:|:-:|:-:|:-:|:-:|
> |EAC+RSL|**97.05**|**93.24**|77.20|78.72|**38.75**|59.88|29.73|84.52|67.80|
> |Ours|96.62|91.91|**79.29**|**83.89**|36.25|**61.11**|**43.24**|**85.20**|**70.33**|
>
> **4.** As EAC does not report the accuracy of each class and the mean accuracy, we could not cite their overall accuracy directly. We re-implemented their method strictly using their code and achieved 89.99% accuracy using their code. However, we noticed that their code is based on ResNet-50, while our Table 1 reports all results of different methods using ResNet-18 as the backbone. To ensure a fair comparison, we modified the pretrained ResNet-50 to pretrained ResNet-18. The best overall accuracy using ResNet-18 reached 88.85%, and the overall accuracy of the last epoch was reported as 88.01%. We believe reporting the best epoch accuracy may be considered cherry-picking, so we reported all the results using the accuracy of the last epoch. We have summarized both the last epoch accuracy and the best epoch overall accuracy in the following table. The results in the table show that our method outperforms EAC under almost all cases. Note that the best overall accuracy does not necessarily mean that the model achieves the best mean accuracy in the same epoch. In fact, the best mean accuracy of our proposed method reached 83.18% under ResNet-18.
>
> |Method|Hap|Neu|Sad|Sur|Dis|Ang|Fea|Overall Acc|Mean Acc|
> |--|:-:|:-:|:-:|:-:|:-:|:-:|:-:|:-:|:-:|
> |EAC last|95.70|86.91|87.87|87.23|59.38|80.25|58.11|88.01|79.35|
> |Ours last|**96.37**|**89.56**|**89.33**|**87.84**|**66.89**|**80.86**|**66.22**|**89.77**|**82.44**|
> |EAC best|96.03|88.68|**90.17**|86.02|61.25|79.63|59.46|88.85|80.18|
> |Ours best|**96.54**|**89.26**|89.75|**88.15**|**65.63**|**81.48**|**64.86**|**89.80**|**82.24**|
>
> **5.** In Fig. 4, the non-overlapping attention maps are from the happy and sad. These classes shouldn't exhibit similar features. However, other classes might be different, where the rationale comes from the definition of compound expressions [2-3]. For instance, the compound expression "fearfully surprised" exists, leading to similar features between these two classes and resulting in overlapping attention maps.
>
> **Limitations:**
>
> We carry out experiments with an extreme case of only 10 samples of both disgust and fear classes. We refer the reviewer to reviewer Ey1e for the summary of training sample numbers and baseline performance due to space limitations. The results illustrate that our method outperforms the SOTA method EAC, mainly in the minor-class accuracy and the mean accuracy. Note that happy (4772 samples) is around 500 times more than the samples of disgust and fear, which leads to low test accuracy for disgust and fear. We think this degree of imbalance might be considered a limitation.
>
> |Method|Hap|Neu|Sad|Sur|Ang|Dis|Fea|Overall Acc|Mean Acc|
> |--|:-:|:-:|:-:|:-:|:-:|:-:|:-:|:-:|:-:|
> |EAC|96.54|**89.71**|**90.79**|**90.58**|87.04|0.00|0.00|85.63|64.95|
> |Ours|**96.79**|**89.71**|90.17|90.27|**87.65**|**7.50**|**1.35**|**86.05**|**66.21**|
>
> [1] The Devil is in the Margin: Margin-based Label Smoothing for Network Calibration. In CVPR.
>
> [2] Sixteen facial expressions occur in similar contexts worldwide. In Nature.
>
> [3] Multi-Label Compound Expression Recognition: C-EXPR Database & Network. In CVPR.

---

> > ### Author Response · Authors · 2023-08-13
> > **Author follow-up**
> >
> > Dear Reviewer N9Q5,
> >
> > We sincerely appreciate the time and effort you dedicated to reviewing our paper.
> >
> > We hope that our responses have effectively addressed your concerns. If you have any additional points of concern, please do not hesitate to bring them to our attention—we would be more than willing to address them. We eagerly await your feedback.
> >
> > Thank you very much!

---

> > > ### Comment · Reviewer_N9Q5 · 2023-08-14
> > >
> > > I am very grateful for the authors' detailed rebuttal and it does address my concern. After improving the paper according to all reviewers, I believe it is a work that has made a good contribution to NeurIPS, so I have changed my rating to "Weak Accept".

---

> > > > ### Author Response · Authors · 2023-08-14
> > > > **Thanks for your time and effort.**
> > > >
> > > > We sincerely appreciate your prompt feedback, as well as the time and effort you invested in reviewing our paper. Your thoughtful and thorough review has significantly contributed to the improvement of our paper's quality. We wholeheartedly value the increased score, and we're thrilled that you have decided to support the acceptance of our paper. It's an honor to have received your responsible and insightful review. Thank you again for your immense contributions to our paper.

---

### Official Review · Reviewer_Ux1i · 2023-07-05

**Soundness:** 4 excellent
**Presentation:** 4 excellent
**Contribution:** 4 excellent
**Rating:** 7
**Confidence:** 5

**Summary:**

This paper focuses on the imbalanced learning problem in facial expression recognition (FER). Unlike existing works in imbalanced learning for image classification, the proposed method addresses imbalanced FER from a novel perspective of label distribution learning. Specifically, the proposed method is motivated by the observation that certain major classes in FER may contain valuable features for the minor classes. Building upon this observation, a re-balanced attention map consistency module is introduced, which encourages the model to extract useful information about the minor classes from both minor and major-class samples through attention map consistency. Additionally, a re-balanced label smooth module is incorporated to leverage prior knowledge of the training data distribution and regularize the classification loss. Experimental results on FER datasets with varying levels of imbalance and different backbone architectures demonstrate the effectiveness of the proposed method, particularly in improving the performance of the minor classes.

**Strengths:**

1. This paper provides an interesting label distribution learning perspective to deal with the imbalanced FER, which is novel to me. It proposes two modules to mine extra information of minor-class samples from both major and minor-class samples. From my understanding, this method is specifically designed for FER and technical novelties are sound.

2. The experimental demonstration well supports the claims. The authors have curated different FER datasets with various imbalance factors and the performance improvement on these imbalanced FER datasets is non-trivial compared with other SOTA methods, especially on the minor classes like fear and disgust. Ablation studies presented in Table 5 indicate that each of the proposed modules contributes to the overall performance. I also noticed that in Table 4, combined with Swin-T, the proposed method achieves remarkable results of 92.31% overall accuracy and 87.71% mean accuracy on the widely used RAF-DB dataset.

3. The proposed method is easy to implement and the paper is written in an easy-to-understand manner.

**Weaknesses:**

1. Lack of experiments. As illustrated by the authors, they utilize the weight (i.e., a hyper-parameter of 0.9999) introduced by Ref. [5] as the re-balanced weight for both the re-balanced attention maps and re-balanced smooth label. However, I think there should be some studies regards of this hyperparameter, and what if we utilize different values for the two proposed modules?

2. Some claims were not supported by experiments. The authors claimed that "EAC employs attention maps from all expression classes, rather than just the labeled class, which is similar to label distribution learning". However, there is a lack of experiments to validate this claim. I believe the authors should include experiments to validate their point of label distribution learning, i.e., what is the performance if we only utilize the attention map of the labeled class instead of all classes?

3. Comparative techniques were not sufficiently performed. For example, what if we use GradCAM instead of CAM to implement attention consistency? What if we utilize the basic label smooth instead of the proposed re-balanced label smooth?


**Questions:**

I have some questions regarding the weight design of the proposed method. Since the weight is utilized by both of the proposed modules simultaneously, I think it is very important for the performance of the proposed method. Therefore, I am curious what the performance would be if we simply set the weight as the inverse of the sample numbers of each class? There should be more discussions regarding the weight design of this method.

**Limitations:**

The limitations of this paper should be discussed. This method is designed specifically for imbalanced learning in FER as FER has the label distribution learning characteristic. However, things might be different in other imbalanced image classification tasks. For example, minor classes in imbalanced CIFAR-100 might contain little common feature with major classes in imbalanced CIFAR-100.

---

> ### Author Rebuttal · Authors · 2023-08-09
>
> We are thrilled to receive your review. We sincerely thank you for acknowledging the strengths of our work.
>
> **Weaknesses:**
>
> **1.** The re-balanced weight is inspired by the effective number of [5]. We visualize the re-balanced weight on the original RAF-DB according to different $\beta$ values. The re-balanced weight is normalized, and the sum of the weight is 1. The standard deviation (Std) can reflect the distribution of the re-balanced weight. From the results in the Table, we can summarize that on  RAF-DB, when $\beta$ = 0.9, the re-balanced weight follows the uniform distribution, while when $\beta$ = 0.9999, the re-balanced weight has a large standard deviation and is similar to the inverse of the sample number. Notice that when $\beta$ = 0.9, our method degrades to EAC plus traditional label smoothing. Thus, EAC plus label smoothing could be viewed as a special form of our method which is suitable for balanced train sets, while our method is a more general form with our introduced RAC and RSL modules.
>
> |$\beta$| Sur        | Fea    | Dis    | Hap    | Sad    | Ang    | Neu    | Std    |
> |----------|------------|--------| -------|  ------|  ------|  ------| -------|------- |
> |0.9| 0.1429     | 0.1429 | 0.1429 | 0.1429 | 0.1429 | 0.1429 | 0.1429 | 0.0000 |
> |0.99| 0.1415     | 0.1505 | 0.1417 | 0.1415 | 0.1415 | 0.1417 | 0.1415 | 0.0031 |
> |0.999| 0.1091     | 0.3227 | 0.1545 | 0.0798 | 0.0917 | 0.1563 | 0.0860 | 0.0789 |
> |0.9999| 0.0959     | 0.4188 | 0.1677 | 0.0306 | 0.0645 | 0.1705 | 0.0520 | 0.1235 |
> |inverse| 0.0939     | 0.4310 | 0.1689 | 0.0254 | 0.0611 | 0.1718 | 0.0480 | 0.1290 |
>
> To solve the imbalanced FER, following [5], we utilize a $\beta$ value of 0.9999 across all our experiments. We also carry out experiments with different $\beta$ values as suggested, and the results are shown below. The results show that using $\beta$ as 0.9999 achieves the best results. Note that simply setting the weight as the inverse of the sample number can also yield good results, as its re-balanced weight is similar to the case when $\beta$ = 0.9999.
>
> |Acc|0.9|0.99|0.999|0.9999|inverse|
> |--|--|--|--|--|--|
> |Overall|88.95|89.50|89.28|**89.77**|89.50|
> |Mean|81.18|81.24|81.75|**82.44**|82.02|
>
> We utilize different values for the two modules as suggested. The first value is for the RAC and the second value is for the RSL. From the results, we observe a trend that with the increase of $\beta$ from 0.9 to 0.9999, both the overall and mean accuracy increase.
>
> |Acc|0.9+0.9999|0.99+0.9999|0.999+0.9999|0.9999+0.9999|0.9999+0.999|0.9999+0.99|0.9999+0.9|
> |--|--|--|--|--|--|--|--|
> |Overall|88.92|89.41|89.57|**89.77**|89.67|89.63|89.47|
> |Mean|81.21|81.68|82.24|**82.44**|82.06|81.93|81.72|
>
> **2.** We carry out experiments with only the attention map of the labeled class instead of all classes to validate our claim of label distribution learning. The results are listed in the following table.
>
> |Method|Hap|Neu|Sad|Sur|Dis|Ang|Fea|Overall Acc|Mean Acc|
> |--|:-:|:-:|:-:|:-:|:-:|:-:|:-:|:-:|:-:|
> |One map|95.02|**90.00**|87.66|85.71|57.50|**80.86**|63.51|88.30|80.04|
> |Ours|**96.37**|89.56|**89.33**|**87.84**|**66.89**|**80.86**|**66.22**|**89.77**|**82.44**|
>
> Our method outperforms the use of only one attention map of the label class on almost all cases. The results validate our claim that FER resembles label distribution learning as utilizing attention map consistency only on the label class fails to learn extra information from other classes on a given sample.
>
> **3.** We conduct experiments utilizing GradCAM instead of CAM to implement attention map consistency. The results are shown in the following table.
>
> |Method|Hap|Neu|Sad|Sur|Dis|Ang|Fea|Overall Acc|Mean Acc|
> |--|:-:|:-:|:-:|:-:|:-:|:-:|:-:|:-:|:-:|
> |GradCAM|**96.46**|87.65|87.03|84.19|59.38|**82.72**|59.46|88.17|79.56|
> |Ours|96.37|**89.56**|**89.33**|**87.84**|**66.89**|80.86|**66.22**|**89.77**|**82.44**|
>
> The performance of utilizing GradCAM is worse than using CAM. As stated in the GradCAM paper, GradCAM contains a ReLU function. We speculate that the ReLU function might diminish some useful information and thus degrades the regularization effect of the attention maps. Furthermore, if we want to get the attention maps of all classes using GradCAM, we need to backward the gradient many times and concatenate the attention map responding to each class, which is not as efficient as using CAM. Based on the better performance and the easier implementation, we choose to use CAM instead of GradCAM in our method.
>
> The basic label smooth means the $\beta$ for the label smooth is 0.9 and the $\beta$ for attention map consistency is 0.9999 based on the above discussion. Thus, the overall accuracy is 89.47% and the mean accuracy is 81.72%, which are lower than our 89.77% and 82.44%, which illustrates the improvement of our RSL upon the basic label smooth.
>
> **Questions:**
>
> Please refer to the 1. in the weaknesses part.
>
> **Limitations:**
>
> Thanks for your suggestion. As our method is specifically designed for FER, it might not work well under datasets without the label distribution learning characteristic, we have added it to the limitation discussion of our revised manuscript.

---

> > ### Comment · Reviewer_Ux1i · 2023-08-14
> > **Thanks for your detailed response.**
> >
> > I've thoroughly reviewed the authors' rebuttal and am generally satisfied with their responses, both to my review and to the other reviews. They have effectively addressed my concerns regarding the re-balanced weight study. The authors provided comprehensive experiments and analyses on the re-balanced weight distribution and the corresponding performance with different hyper-parameters. The comparison with the one-class attention map reveals that there is indeed extra information from attention maps besides the target class.
> > Overall, I recognize several merits of this paper:
> > 1. The idea of extracting additional knowledge regarding minor classes from both major and minor class samples is novel.
> > 2. The experimental results on RAF-DB using Swin-T as the backbone are highly promising.
> > 3. In the rebuttal, the authors presented results on AffectNet and the extremely imbalanced RAF-DB. Considering the accuracy of minor classes, it's clear that the method is well-suited for the imbalanced FER task.
> >
> > To the best of my knowledge, there is no similar method that extracts minor-class information from both the major and minor-class samples to effectively address the imbalanced FER task. The motivation and idea are novel and interesting to me. Given the method's strong performance, achieving state-of-the-art results in the imbalanced FER task, I believe this paper deserves acceptance. Therefore, I recommend accepting this paper.

---

> > > ### Author Response · Authors · 2023-08-14
> > > **Thanks for your feedback.**
> > >
> > > We wholeheartedly thank you for dedicating time to thoroughly read all the reviews and responses. The merits you highlighted align seamlessly with the strengths of our paper. Your recognition of the novelty in our motivation and method is greatly appreciated. We're thrilled to hear that you believe our paper deserves acceptance. Thank you very much for your support.

---

### Official Review · Reviewer_31BV · 2023-07-05

**Soundness:** 4 excellent
**Presentation:** 4 excellent
**Contribution:** 3 good
**Rating:** 7
**Confidence:** 4

**Summary:**


In the paper, the authors tackle the issue of imbalanced learning in the facial expression recognition task by introducing a fresh approach. Their proposed method revolves around the concept of attention consistency under spatial transforms, aiming to extract information about multiple classes from each sample. This approach effectively enhances the performance of under-represented classes. The authors evaluate their approach on two datasets and successfully achieve state-of-the-art performance with a transformer backbone.

**Strengths:**

(+) They achieve state-of-the-art (SOTA) performance on the RAF-DB dataset with the Swin backbone.

(+) Table 4 clearly demonstrates the performance improvement brought by the proposed approach. It highlights the significant increase in performance for both minor classes and the overall score.

(+) Despite attention consistency being proposed in previous work, the authors cleverly and novelly employ a modified version of it incorporating rebalanced smooth labeling. They provide a reasonable hypothesis, stating that information for minor classes can be captured from all training samples.

(+) The visualization results presented in both section 4.7 and the appendix provide further evidence that the proposed approach functions effectively.

**Weaknesses:**

(-) To further support the idea, I suggest conducting evaluations on multi-label classification using other facial expression recognition (FER) datasets, such as BP4D. If the hypothesis is correct, the proposed approach should also work effectively in a different type of FER task. Testing it in this manner would provide a good opportunity for validation.

(-) The two methods proposed in the paper, in terms of their structure, are not entirely novel. However, there is potential for further refinement and innovation in their design.

**Questions:**

- What is the purpose of utilizing Global Average Pooling to extract features? Doesn't it result in a loss of information?

- Based on the findings presented in Table 4, it appears that the performance improvement for minor classes surpasses that of other classes. Regarding the fairness aspect (although it may not be the primary focus of the paper), what can be said about the trained models? Do you believe that your method also facilitates the development of fairer models?

**Limitations:**

Given that it is a facial expression recognition task, privacy could potentially be a concern.

---

> ### Author Rebuttal · Authors · 2023-08-09
>
> We sincerely appreciate your insightful review. The strengths you highlighted align perfectly with the core contributions of our paper, and they indeed encapsulate what we take great pride in. Your review is undoubtedly one of the most exhilarating ones we have received in over a year. Thank you very much.
>
> **Weaknesses:**
>
> **1.** Thanks for your valuable suggestion. Though we wanted to carry out experiments on BP4D, we emailed the authors of BP4D and they told us they could not enter into an agreement for the dataset at this time and we could not download it. As suggested, we conduct experiments on the multi-label FER task. We utilize a widely used multi-label FER dataset EmotioNet [1, 2], which is published in CVPR and provides 6 basic expressions and 10 compound expressions. There are a total of 2,478 images and we utilize 80% of them for training and the rest for testing. The results of different methods are summarized below.
>
> |Method | Hap    | Hap Dis | Sad Dis | Sad   | Hap Sur  | Ang   | Sur  | Ang Sur  | Awe     |
> |-------| :-:    | :-:     |  :-:    | :-:   |    :-:   | :-:   | :-:  |  :-:     | :-:     |
> |SCN     | 82.24 |**71.74**|**80.00**|**66.67**| 0.00   | 35.71 | 30.77| 15.38    |**61.54**|
> |BBN     | 77.63 | 69.57   | 65.71   | 62.96 | 35.29    |**64.29**| 53.85| 7.69     | 53.85   |
> |RUL     | 75.66 | 62.32   | 62.86   | 59.26 | 35.29    |50.00  |**69.23**|46.15|53.85 |
> |EAC     | 78.29 | 69.57   | 68.57   |**66.67**|**70.59**| 57.14 | 46.15 | 46.15    | 46.15   |
> |Ours    |**86.18**| 65.94 | 68.57   | 55.56 |**70.59**| 57.14 | 46.15   |**53.85**| 46.15   |
>
>
> |Method | Dis     | Ang Dis | App   | Fea Ang  | Sad Ang | Fea  | Fea Sur| Overall Acc| Mean Acc|
> |-------| :-:    | :-:     |  :-:    | :-:   |    :-:   | :-:     | :-:  |  :-:     |:-:     |
> |SCN      | 8.33    | 0.00    | 0.00  |**30.00**|  0.00   |  0.00| 0.00 |  59.43   |  30.15 |
> |BBN      | 8.33    | 18.18   | 0.00  |**30.00**|**30.00**| 50.00| 0.00|  60.45   |  39.21 |
> |RUL      | 50.00   |**45.45**| 0.00  |**30.00**| 20.00   | 30.00| 0.00 |  59.43   |  43.13 |
> |EAC      |**66.67**| 27.27   | 0.00  |**30.00**| 20.00 | 40.00|**50.00**|  64.71   |  48.95 |
> |Ours     |**66.67**| 36.36   |**10.00**|**30.00**|**30.00**|**60.00**|**50.00**|**66.73**|  **52.07**|
>
> We have organized the 16 different expression classes in [1, 2] into two tables based on the number of their training samples, arranged from largest to smallest.  There are instances of similar test accuracy scores when the test sample numbers for certain compound expressions are small, which can be attributed to the challenges associated with collecting samples for compound expressions.
>
> The results demonstrate that our method consistently achieves the best overall and mean accuracy in the context of the multi-label FER task. Additionally, our method excels in terms of performance within the minor classes in the second table. The multi-label FER task is also aligned with our motivation, as the definition of compound expressions implies that a sample could encompass features from multiple basic expression classes. For instance, a sample in the fearfully angry (Fea Ang) class contains features from both the fear and angry classes.
>
> **2.** Thanks for bringing that up. Though the structure of our RAC and RSL modules is not entirely novel, their motivation is. We propose RAC to guide the model to mine extra information related to minor classes from all training samples for the first time. RSL acts as a complementary module for RAC and utilizes the extra information regarding the label distribution of the imbalanced training data to weigh minor classes more heavily. Both of the two modules learn existing extra knowledge to improve the imbalanced learning of FER, which aligns with the title of our paper.
>
> **Questions:**
>
> **1.** The features used for attention map consistency are extracted before the Global Average Pooling (GAP) layer to prevent information loss, as we employ F (features before GAP) instead of f (features after GAP) in Eq.(2) of our paper. The success of our method hinges on preserving the spatial information related to expression features since the attention map consistency also extends across the spatial dimensions.
>
> **2.** Yes, we believe our method achieves a fairer FER model. Though in Table 4, the performance improvement for minor classes sometimes surpasses that of other classes, the test accuracy of minor classes is still lower than the major classes. We believe a fair FER model should achieve similar accuracy on different expression classes. Based on the results in Table 4 and the results on AffectNet in the rebuttal for reviewer Ey1e, the gap between the accuracy of major and minor classes is distinctly smaller using our method. Thus, we believe our method achieves a fairer FER model.
>
> **Limitations:**
>
> Thanks for your suggestion. We have added the limitation discussion in our revised manuscript. As there might be a privacy problem in FER, we could further combine our method with the technology of differential privacy [3] to help preserve privacy during FER.
>
> [1] Emotionet: An accurate, real-time algorithm for the automatic annotation of a million facial expressions in the wild.
>
> [2] Emotionet challenge: Recognition of facial expressions of emotion in the wild.
>
> [3] Privacy Preserving Face Recognition Utilizing Differential Privacy.

---

> > ### Comment · Reviewer_31BV · 2023-08-14
> > **Response to the rebuttal**
> >
> > I appreciate your comprehensive feedback. As I comprehend your comments, it appears that you utilized the EmotioNet dataset. However, it seems that the outcomes are coming from a 'classification' problem, rather than a multi-label classification task where each instance could be linked to multiple labels concurrently. To illustrate, if the AU units are accessible within that dataset, could you kindly share the results involving a multi-label classification task?
> >
> > I am inclined to support the acceptance of this paper, as the authors have adeptly addressed both my concerns and those of fellow reviewers. Through their thorough experiments, as presented in both their response and the manuscript itself, they have convincingly demonstrated their model's advancement to state-of-the-art status.

---

> > > ### Author Response · Authors · 2023-08-16
> > > **Thanks for your time and effort.**
> > >
> > > We extend our heartfelt gratitude for your valuable time and dedication to reviewing our paper.
> > >
> > > We are truly delighted by your endorsement of our paper's acceptance and your commitment to thoroughly peruse both the reviews from other reviewers and all of our comments. Your professional and insightful review is a significant honor for us, and we deeply appreciate it.
> > >
> > > In response to your suggestion, we have incorporated the AU recognition results into the tables provided below. In our experiments, we changed the loss to binary cross-entropy loss and added a sigmoid layer after the FC layer. During the test phase, logits larger than 0.5 are classified as 1, while logits smaller than 0.5 are classified as 0. Following the tradition of AU recognition, we report the F1 score for each action unit and calculate the average F1 score across all classes. As the AU labels are also imbalanced, we have organized the AUs in descending order based on the number of labels in each class.
> > >
> > > |Method|AU 25|AU 12|AU 10|AU 4|AU 1|AU 2|AU 15|
> > > |------|:-: |:-:  | :-: | :-: | :-: | :-:| :-: |
> > > |Baseline|**0.9799**|0.9935|0.7251|**0.9291**|0.8522|0.7391|0.6885|
> > > |SCN|0.9759|**0.9951**|0.7225|0.9258|0.8364|0.7692|**0.7164**|
> > > |RUL|0.9564|0.9754|0.6633|0.8767|0.7302|0.6549|0.3714|
> > > |EAC|0.9757| 0.9903|0.7188|**0.9291**|0.8689|0.8317|0.7013|
> > > |Ours|0.9733|0.9919|**0.7353**|0.9214|**0.8926**|**0.8515**|0.7042|
> > >
> > > |Method|AU 20 |AU 26|AU 7|AU 17|AU 5|AU 9|AU 24|Avg.|
> > > |------|:-: |:-:  | :-: | :-: | :-: | :-:| :-: | :-:|
> > > |Baseline|0.5306|0.4615|0.2353|0.3636|0.5405|0.4516|0.3810|0.6337|
> > > |SCN|**0.5714**|0.3784|0.2941|0.4118|0.5556|0.3871|0.3158|0.6325|
> > > |RUL|0.2373|0.1333|0.0513|0.2326|0.3448|0.1463|0.0000|0.4553|
> > > |EAC|0.5652|0.4878|0.3684|**0.4444**|**0.5854**|0.4516|0.4167|0.6668|
> > > |Ours|**0.5714**|**0.5581**|**0.4324**|0.3889|0.5500|**0.5294**|**0.4545**|**0.6825**|
> > >
> > >
> > >
> > >
> > >
> > > From the results, we can observe that our method achieves the best average F1 score compared to other FER methods. Furthermore, our method attains the highest F1 score on AU 9 and AU 24 with the least training labels, which is consistent with our experimental results in expression recognition.
> > >
> > > We recognize that RUL is not well-suited for multi-label classification and results in low performance. The reason behind this lies in the fact that RUL compares two samples from different expression classes to learn uncertainty. However, when a sample possesses multiple labels concurrently, such a comparison becomes less meaningful. EAC and our method perform well since the main component of these two methods is attention map consistency [1], which was originally proposed for multi-label classification. Notably, our method outperforms EAC as we re-weight the attention map and integrate the information of the label distribution, making it more suitable for handling imbalanced tasks.
> > >
> > > We have included this experiment along with its corresponding discussion in our revised manuscript, and we firmly believe that this addition will enhance the credibility of our method. We sincerely appreciate your valuable suggestion.
> > >
> > > If we have addressed your concerns, please kindly let us know. If you have any other additional concerns, please do not hesitate to bring them up. We would try our best to address them. Thank you very much for your immense contribution for reviewing our paper!
> > >
> > >
> > > [1] Visual Attention Consistency under Image Transforms for Multi-Label Image Classification. In CVPR 2019.

---

> > > > ### Comment · Reviewer_31BV · 2023-08-21
> > > >
> > > > I appreciate your response, and given the results of these experiments (which they also have shown the results for multi-label classification in the rebuttal) and novelties described by the authors, that have shown to reach the state-of-the-art approach to the problem, I recommend acceptance of this paper to the conference.

---

> > > > > ### Author Response · Authors · 2023-08-21
> > > > >
> > > > > Thank you for your feedback and for dedicating your time and effort to reviewing our paper. Your valuable suggestions have significantly contributed to enhancing our work. We deeply appreciate your support for our paper.

---

### Official Review · Reviewer_Ey1e · 2023-07-05

**Soundness:** 2 fair
**Presentation:** 3 good
**Contribution:** 3 good
**Rating:** 6
**Confidence:** 5

**Summary:**

The paper presents an approach to tackle the imbalance problem in facial expression recognition. The approach makes use of information from the majority class to help improve performance of the minority classes. The proposed approach is evaluated on two public FER datasets. Different backbone networks are evaluated, along with comparisons to state of the art and ablation studies.

**Strengths:**

The paper investigates a challenging problem. and the proposed approach using attention map consistency is interesting. Encouraging results are shown on two public datasets. Ablation study is generally well conducted.

**Weaknesses:**

Experimental design is lacking. FER2013 is an old dataset; newer FER datasets should be evalauted (e.g., AffectNet - Mollahosseini, Ali, Behzad Hasani, and Mohammad H. Mahoor. "Affectnet: A database for facial expression, valence, and arousal computing in the wild." IEEE Transactions on Affective Computing 10.1 (2017): 18-31.). Along with being newer, AffectNet needs to be evaluated as the approach is on imbalanced FER datasets. The paper conducts experiments under different imbalance factors, however, AffectNet is one of the largest and most imbalanced datasets available for FER. For example, in AffectNet, Happy is ~46% of the data and contempt is ~1% of the data. How does the proposed approach handle these extreme imbalance issues?

The comparison to CB [5] in Table 1 is not clear. This paper is not on FER and does not evaluate RAF-DB. What is this?

Paper states the accuracy of each class is not commonly shown in FER papers. This is not accurate. If it is not in a table, this is often shown through confusion matrices (e.g., Farzaneh, Amir Hossein, and Xiaojun Qi. "Facial expression recognition in the wild via deep attentive center loss." Proceedings of the IEEE/CVF winter conference on applications of computer vision. 2021.)

**Questions:**

How does the proposed approach handle extreme imbalance such as in AffectNet (e.g., Happy vs. Contempt)?

What is reference [5] in Table 1? This is not a FER paper that is being compared to.

**Limitations:**

The limitations of the proposed work are not addressed. How does the approach work on extremely imbalanced data? What kind of societal impact can this have? There are always ethical concerns with FER systems.

---

> ### Author Rebuttal · Authors · 2023-08-09
>
> Thank you very much for your insightful review, which has assisted us in enhancing the quality of our paper.
>
> **Weaknesses:**
>
> **1.** Thanks for your valuable suggestion. We have conducted experiments on AffectNet with both 7 and 8 expression classes. Due to the time consumption, we use ResNet-18 as the backbone for all methods. As AffectNet has a balanced test set with 500 samples of each class, the mean accuracy equals the overall accuracy.
>
> |Method | Hap        | Neu       | Sad      | Ang       | Sur      | Fea     | Dis      | Mean Acc  | Overall Acc|
> |-------| :-:        | :-:       |  :-:     |   :-:     |   :-:    |   :-:   |  :-:     |  :-:      | :-:        |
> | SCN   | **95.20**  |**82.70**  | 44.20    | 56.30     | 35.80    | 38.00   | 20.90    | 53.30     | 53.30      |
> | BBN   |  87.00     | 57.10     | **66.80**| 58.30     | 54.90    | 71.10   | 30.10    | 60.76     | 60.76      |
> | RUL   | 90.50      | 62.40     | 64.70    | **69.30** | 60.80    | 49.00   | 34.20    |  61.56      | 61.56      |
> | EAC   | 91.40      | 64.50     | 65.70    | 66.30     |**61.60** | 60.90   | 45.80    | 65.17     | 65.17      |
> |Ours   | 86.20      | 59.00     | 64.20    | 66.50     | 57.80    |**64.50**|**61.90** | **65.73** |**65.73**   |
>
>
> |Method | Hap        | Neu       | Sad     | Ang       | Sur      | Fea     | Dis     | Con     | Mean Acc  |Overall Acc|
> |-------| :-:        | :-:       |  :-:    |   :-:     |   :-:    |   :-:   |  :-:    | :-:     | :-:       | :-:       |
> | SCN   | **94.60**  | **74.90** | 58.20   | 63.80     | 40.90    | 43.20   | 30.80   |2.20     |  51.08    | 51.08     |
> | BBN   | 78.40      | 58.40     | 60.60   | **67.70** | 59.40    | 55.00   | 37.00   |46.70    |  57.90    | 57.90     |
> | RUL   | 71.00      | 63.40     | 46.60   | 54.90     | 53.70    | 58.60   | 44.70   |47.70    |  55.08    |  55.08    |
> | EAC   | 84.00      | 58.80     |**65.00**| 65.90     | **62.20**| 60.30   | 46.10   |41.90    | 60.53     | 60.53     |
> |Ours   | 78.60      | 54.30     | 63.80   | 59.50     | 57.60    |**64.10**|**59.40**|**60.00**| **62.16** |**62.16**  |
>
> From the results, we could draw the conclusion that our method achieves the best mean and overall accuracy on the test set under both 7 or 8 classes. Besides, we notice that our method improves existing methods on the minor classes of fear (Fea), disgust (Dis), contempt (Con) by remarkable margins, which illustrates that our method is more suitable for the imbalanced learning of FER task. Our method even achieves 60.00% accuracy on the contempt class under 8 classes. Furthermore, though "Happy is ~46% of the data, and contempt is ~1% of the data," our method clearly decreases the test accuracy gap between happy (78.60%) and contempt (60.00%) compared with other methods under 8 classes, which means our method is fairer and achieves a more balanced test accuracy.
>
> **2.** The re-balanced weight design of our method is inspired by CB [5], which is a widely used method in imbalanced learning of image classification. We carry out experiments using [5] to show that simply weighing the cross-entropy loss cannot bring much improvement in the FER task, which illustrates the superiority of our method using re-balanced attention map consistency to mine extra minor-class information from all FER samples.
>
> **3.** We report the accuracy of each class and arrange them according to the sample number of each class instead of plotting confusion matrices due to space limitations and visualization simplicity. As suggested, we display the main results of our paper using confusion matrices in the uploaded PDF in the above "global response" for further reference.
>
> **Questions:**
>
> Please refer to 1. and 2. in the weaknesses part.
>
> **Limitations:**
>
> Thanks for your valuable suggestion. We study our method under extremely imbalanced data. For example, on RAF-DB, we only keep 10 training samples for both fear and disgust and keep other classes the same, leading to a extremely imbalanced train set summarized as below.
>
> |Class  | Hap   | Neu   | Sad     | Sur    | Ang   | Dis    | Fea  |
> |------ | :-:   | :-:   |  :-:    |   :-:  |   :-: |   :-:  |  :-: |
> |Number |4772   | 2524  | 1982    | 1290   | 705   | 10     | 10   |
>
> The test accuracy on the original test set of RAF-DB is shown below.
>
> |Method   | Hap   | Neu   | Sad  | Sur     | Ang   | Dis    | Fea  | Overall Acc| Mean Acc|
> |------   | :-:   | :-:   |  :-: |   :-:   |   :-: |   :-:  |  :-: |  :-:       |  :-:    |
> |Baseline |96.03|**91.03**|87.66 | 88.15   |82.10  | 0.00   |0.00  |84.71       |   63.57 |
> |EAC      |96.54 |89.71|**90.79**|**90.58**|87.04  | 0.00   |0.00  |85.63       |   64.95 |
> |Ours     |**96.79**|89.71|90.17 | 90.27   |**87.65**|**7.50**|**1.35**|**86.05**|**66.21**|
>
> The results illustrate that our method still outperforms the SOTA method EAC on extremely imbalanced data, mainly in the minor classes accuracy and the mean accuracy. However, the extreme case with happy having 4772 samples, which is around 500 times more than the samples for disgust and fear (only 10 samples each), also leads to a very low test accuracy of our method for disgust and fear. This should be considered a limitation of our method. As for the ethical concerns, we could further combine our method with the technology of differential privacy [1] to help preserve the privacy during recognition.
>
> [1] Privacy Preserving Face Recognition Utilizing Differential Privacy.

---

> > ### Comment · Reviewer_Ey1e · 2023-08-11
> >
> > Thank you for the response to my concerns. I have changed my rating to weak accept based on these new experiments.

---

> > > ### Author Response · Authors · 2023-08-11
> > > **Response to Reviewer Ey1e**
> > >
> > > Thank you very much for providing a prompt response. Your review has been immensely helpful in improving the quality of our paper. We are truly appreciative of the increased score.

---

### Author Rebuttal · Authors · 2023-08-10

We sincerely thank all reviewers for their time and effort in reviewing our paper, as well as for their insightful feedback!

We are encouraged that Reviewers 31BV, Ux1i, and ss3S all find our main idea of mining extra knowledge about minor classes from training samples of both major and minor classes to be novel. Reviewer 31BV thinks that we "cleverly and innovatively" designed our method, and Reviewer Ux1i also believes that our method is specifically designed for FER and the technical innovations are well-founded. Reviewers Ey1e and N9Q5 assert that we are addressing a challenging problem, and that our method is interesting and holds promise in solving imbalanced FER through a re-balanced strategy. Moreover, Reviewers ss3S, Ux1i, and 31BV all concur that our method achieved state-of-the-art performance on the RAF-DB dataset with the Swin backbone, and our results "clearly indicate that the proposed method outperforms competing methods" under the imbalanced learning FER task, particularly in terms of the accuracy of minor classes. Reviewers Ux1i and N9Q5 also note the thoroughness of the experiments in our paper. Reviewers ss3S and Ux1i remark that Table 4 clearly demonstrates the performance improvement brought about by the proposed approach, and that our paper is well-written.


Addressing the weaknesses of our paper, Reviewer Ey1e suggested that we add experiments on AffectNet. Following this suggestion, we conducted experiments and found that our method also outperforms other FER methods on this extremely imbalanced dataset. Reviewer 31BV suggested evaluating our method under multi-label FER task. We conducted experiments as suggested and found that our method still performs well. Multi-label FER task aligns with our statement that a sample might contain features from several expression classes. For instance, the class "fearfully angry" combines features from both the fear and angry classes. Reviewer Ux1i primarily raised concerns about the hyper-parameter study, prompting us to add the corresponding experiments. Reviewer N9Q5 pointed out some unclear descriptions regarding our proposed modules, so we added more detailed discussions of our RAC and RSL modules. We greatly appreciate the extensive feedback provided by Reviewer ss3S, which has significantly aided in improving the quality of our paper. Additionally, Reviewer ss3S suggested exploring other transformations, leading us to include corresponding experiments as suggested.

We address specific questions below and respond to each review with a separate response in detail. We have incorporated all feedback into the revised version of our manuscript and supplementary material. With our responses, we hope that we have addressed the reviewers' concerns. Please let us know if there are any further questions or comments. We are eager to do our utmost to address them.

---

### Decision · Program_Chairs · 2023-09-21

**Decision:**

Accept (poster)

**Comment:**

While attention consistency and label smoothing (Hinton et al.) have been widely studied in the literature (e.g. in the EAC work), this paper has introduced some novel modifications, such as a re-balanced weight based on the effective number in [5]. The modified techniques, when combined as a whole, performs effectively for facial expression recognition tasks on both major and minor classes. All reviewers are positive on this submission. The AC agrees with them and suggests accepting this submission.